# Thrombopoietin increases susceptibility for EVI1 + KMT2A-MLLT3-driven AML expressing stem cell genes linked to poor outcome

Hugues-Étienne Châtel-Soulet[1,10], Sabine Juge[1,10], Ana Luisa Pereira[2], Jonathan Seguin [1,3], Athimed El Taher [3,4], Federica Valigi [1], Zivojin Jevtic[1], Rathick Sivalingam[1], Frederik Otzen Bagger [1,5], Paul Büschl [2], Marwa Almosailleakh[1,6], Alexander Tzankov [7], Wei Tong [8], Mineo Kurokawa[9], César Nombela Arrieta [2] & Juerg Schwaller [1]✉

To address the cellular origin of ecotropic virus integration site 1 (EVI1)-expressing aggressive KMT2A-rearranged acute myeloid leukaemia (AML) we integrate an Evi1-GFP reporter allele in the inducible iKMT2A-MLLT3 mouse model. We observe that a single injection of thrombopoietin (TPO) selectively increases the number of cycling Evi1⁺ haematopoietic stem cells (HSC) and accelerates AML initiation. Comparison of mouse Evi1⁺ KMT2-MLLT3⁺ AML originating from TPO-stimulated HSC with human EVI1⁺AML reveals higher expression of HSC genes including IL12Rβ2 and INPP4B linked to poor disease outcome of patients of four large AML cohorts. Knockdown experiments show exclusive MECOM-dependency of human EVI1^high KMT2A-rearranged OCI-AML4 cells while reduction of IL12Rβ2 also impairs clonogenic growth of EVI1^low MOLM-13, THP-1 or HL-60 AML cells. Collectively, we show that exogenous factors like TPO can increase the susceptibility for iKMT2A-MLLT3-driven HSC-originating Evi1⁺ AML expressing stem cell genes linked to transformation maintenance of cell lines, and poor disease outcome of patients.

Acute myeloid leukaemia (AML) is a genetically heterogeneous disease characterized by uncontrolled accumulation of myeloid progenitor cells. Up to 40% of the patients carry chromosomal translocations associated with the expression of fusion genes that often involve transcriptional co-regulators such as the lysine methyltransferase 2 A (KMT2A, aka MLL1)[1,2]. The cellular origin of KMT2A-r AML remains a topic of ongoing debates. Functional studies in mice have shown that many transcriptionally active AML-associated fusion genes aberrantly induce or maintain stem cell properties in haematopoietic stem and progenitor cells (HSPC). Some of the most prevalent KMT2A fusions including KMT2A-MLLT3 (aka MLL-AF9) or KMT2A-MLLT1 (aka MLL-ENL) were shown to have leukaemia-inducing activity at multiple levels

[1]University Children's Hospital & Department of Biomedicine, University of Basel, Basel, Switzerland. [2]Department of Medical Oncology and Haematology, University Hospital Zürich, Zürich, Switzerland. [3]Swiss Institute of Bioinformatics (SIB), Basel, Switzerland. [4]Bioinformatics Core Facility, University of Basel, Basel, Switzerland. [5]Center for Genomic Medicine, Rigshospitalet, University of Copenhagen, Copenhagen, Denmark. [6]Biotech Research and Innovation Center (BRIC), Faculty of Health and Medical Sciences, University of Copenhagen, Copenhagen, Denmark. [7]Institute for Pathology, University of Basel, Basel, Switzerland. [8]Division of Haematology, Children's Hospital of Philadelphia, Philadelphia, PA, USA. [9]Department of Haematology and Oncology, Graduate School of Medicine, University of Tokyo, Tokyo, Japan. [10]These authors contributed equally: Hugues-Étienne Châtel-Soulet, Sabine Juge. ✉e-mail: J.Schwaller@unibas.ch

 

of the haematopoietic hierarchy resulting in either lymphoid, myeloid or mixed-lineage leukemia[3]. Inherently, fast cell cycle progression was proposed to make granulocyte-macrophage progenitors (GMP) particularly permissive for transformation by retrovirally expressed rKMT2A-MLLT3[4]. GMP-derived rKMT2A-fusion leukemic stem cells ectopically express a limited stem cell program while maintaining the identity of the cell of origin[5]. Retroviral rKMT2A-MLLT3 expression in lineage marker-depleted (Lin⁻), Sca-1⁺Kit⁺ (LSK) cells resulted in a more aggressive disease than in more committed GMP[6,7]. Bone marrow (BM) reconstitution with LSK revealed that rKMT2A-MLLT3 can generate a more aggressive disease initiating within the HSC compartment associated with expression of the transcription factor ecotropic virus integration site 1 (Evi1)[6]. Similarly, experiments with inducible transgenic mouse models suggested different susceptibility of distinct cells from the haematopoietic hierarchy for transformation by KMT2A fusions[8–10]. In particular, expression of the iKMT2A-MLLT3 fusion in long-term haematopoietic stem cells (LT-HSC) transplanted into irradiated recipients induced a particularly rapid and invasive disease phenotype characterized by high Evi1 expression only in a small fraction of mice, however, the underlying mechanism remained unclear[9].

The MDS1 and EVI1 complex (MECOM) gene locus at 3q26 is target of recurrent AML-associated chromosomal alterations that result in deregulated EVI1 expression[11]. However, EVI1 expression is also found in a significant fraction of paediatric and adult AML independently of any 3q26 rearrangements, and seen as marker of poor outcome[12–14]. While some studies suggested that EVI1 expression is induced by KMT2A fusions, others proposed that the fusions rather maintain higher EVI1 levels linked to the cell of origin[15]. Characterization of a transgenic reporter mouse line, in which an IRES-GFP cassette is knocked into the endogenous Mecom gene locus indicated that Evi1 expression mostly marks hematopoietic cells with long-term multilineage repopulating activity[16]. Targeting of a GFP cassette to the transcriptional start site of the Mds1 locus confirmed Evi1 expression in the LT- and short-term (ST) HSC compartments[17].

In this work, we introduce the Evi1-IRES-GFP reporter allele in the iKMT2A-MLLT3 mouse line ("KME" mice) to model the origin of EVI1⁺ KMT2A-rearranged AML. We find that in contrast to 5-fluorouracil (5-FU) or polyinosinic:polycytidylic acid (poly(I:C)) stimulation, short-term in vivo exposure to recombinant thrombopoietin (TPO) selectively and significantly increases the fraction of cycling Evi1⁺ LT-HSC and multipotent progenitor 1 (MPP1) cells that induce an aggressive Evi1⁺ iKMT2A-MLLT3-driven AML upon transplant. Comparison with expression signatures from MECOM⁺ AML from four large AML patient cohorts reveals common differential expression of several HSC genes, including IL12Rβ2. Notably, knockdown (KD) of MECOM, and IL12Rβ2 significantly impairs the clonogenic growth of human KMT2A-rearranged OCI-AML4 AML cells expressing high MECOM levels. Interestingly, KD of IL12Rβ2 also impairs the clonogenic growth of EVI1low AML cell lines (MOLM-13, THP-1 or HL-60). Collectively, our work suggests that exogenous factors like TPO can increase the susceptibility of EVI1⁺ HSC for malignant transformation by KMT2A fusion oncogenes like KMT2A-MLLT3, resulting in aggressive AML expressing distinct HSC genes that may serve as biomarkers and/or future therapeutic targets.

## Results

### TPO but not 5-FU or poly-(I:C) increases the number of Evi1⁺ HSC

Previous studies found that systemic viral infection increased Evi1 expression in murine LT-HSC[18]. We therefore wondered whether exogenous stimuli may modulate Evi1 levels in the HSPC compartment and influence iKMT2A-MLLT3-induced AML. We explored factors known to affect the number of murine HSPC, including 5-FU, poly(I:C) and TPO. 5-FU impairs DNA replication resulting in cytotoxic death of cycling haematopoietic progenitors, poly(I:C) mimics an infection-mediated

interferon response activating dormant HSC in vivo, and TPO regulates HSC quiescence and interaction with the BM niche[19–25].

To address the effects of these exogenous stimuli on Evi1⁺ HSPC, we injected the respective compounds into Evi1GFP/⁺ mice and analysed BM cells after 24 hours ((poly(I:C), TPO), or 3 and 6 days (5-FU) respectively (Supplementary Fig. 1A). In accordance with previous observations, we found that compared to PBS, a single injection of 5-FU (i.p., 150 mg/kg body weight (BW)) resulted in a significant reduction of white blood cells (WBC) (3 d: −31.1%, adj.$p = 0.013$; 6 d: −40.5%, adj.$p = 0.0007$), platelet (PLT) counts (3 d: −15.7%, n.s.; 6 d: −53.6%, adj.p < 0.0001) and BM cellularity (3 d: −54.9%, adj.$p = 0.0001$; 6 d: −78.8%, adj.p < 0.0001) in wildtype mice (Supplementary Fig. 1B–D). Although the number of MPP1 did not change, MPP2 and MPP3 cells decreased after 3 days (MPP2: −87.2%, adj.p < 0.0001; MPP3: −95.5%, adj.p < 0.0001), while numbers of LT-HSC increased significantly 6 days after injection (+76.3%, adj.p < 0.013) (Fig. 1A–B; Supplementary Fig. 1E–F). 5-FU treatment also resulted in expansion of Evi1neg cells in all cellular fractions (Fig. 1C–D; Supplementary Fig. 1G–H). A single injection of poly(I:C) (i.p., 5 mg/kg BW) decreased (vs. PBS) WBC (−39.5%, adj.$p = 0.0003$) and BM cellularity (−37.5%, adj.$p = 0.0196$) (Supplementary Fig. 1A, C) accompanied by a significant increase of LT-HSC (+74,9%, adj.$p = 0.0003$), MPP2 (+306.8%, adj.$p = 0.0068$) and MPP3 (+280.0%, adj.$p = 0.016$), extending previous observations[21] (Fig. 1A; Supplementary Fig. 1E–F). Similar to 5-FU, exposure to poly(I:C) resulted in a reduction of the Evi1high LT-HSC fraction (PBS: 23.4 ± 2.7, pI:pC: 8.176 ± 2.6, in % of LT-HSC, n.s.) with no changes in the more differentiated cell populations (Fig. 1C–D; Supplementary Fig. 1G–H).

By contrast, compared to PBS, a single dose of TPO (i.p., 200 mg/kg BW) almost doubled the number of LT-HSC (579 ± 55.6 vs 1165 ± 175.4 per million Lin⁻ cells, adj.$p = 0.0042$) in the BM after 48 h, associated with a slight increase of WBC (+6.6%, n.s.) and a trend to a higher overall BM cellularity (+6.2%, n.s.) (Fig. 1A; Supplementary Fig. 1B–D). Most importantly, TPO significantly increased the number of Evi1high LT-HSC (PBS: 23.4 ± 2.7, TPO: 50.8 ± 2.2, in % of LT-HSC, adj.p < 0.0001), MPP1 (PBS: 20.9 ± 3.4, TPO: 47.0 ± 2.2, adj.p < 0.0001) and MPP2 (PBS: 6.0 ± 1.7, TPO: 22.9 ± 1.9, adj.$p = 0.0014$), while no significant changes were seen in MPP3, MPP4 (Fig. 1C–D; Supplementary Fig. 1G–H). Similarly, a single dose of the TPO receptor agonist Romiplostim ("RP", 200 mg/kg BW, i.p.) increased LT-HSC, MPP2 and MPP3 (Fig. 1A; Supplementary Fig. 1D–E). RP also significantly increased the fraction of Evi1high LT-HSC (PBS: 23.4 ± 2.7, RP: 46.5 ± 3.9, in % of LT-HSC, adj.$p = 0.0007$) and -MPP1 (PBS: 20.9 ± 3.4, RP: 48.3 ± 2.8, adj.p < 0.0001) (Fig. 1C–D). Concurrently, whole-bone high-resolution imaging[26] revealed more GFP⁺/Kit⁺ (= Evi1⁺ HSPC) cells in the BM from TPO-treated KME mice compared to PBS-treated controls (Fig. 1E–G). Collectively, we found that a single injection of TPO or the synthetic TPO receptor agonist RP, but not of 5-FU or poly(I:C), increases the number Evi1-expressing LT-HSC and MPP1/2 in mice.

### TPO exposure increases cycling of Evi1high LT-HSC and MPP1 cell fractions

As TPO regulates LT-HSC cell cycle progression[22–24], we explored the cycle dynamics of Evi1-expressing HSPC. As expected, at steady-state in untreated animals, the LT-HSC were less cycling than MPP1 reflected by their S/G2/M fraction (10.5% ± 0.25 vs. 14.4% ± 0.7, $n = 2$) (Supplementary Fig. 1I). TPO-treated animals exhibited a more activated cycling profile in the LT-HSC compartment, with an increased S/G2/M fraction similar to the one seen in MPP1 at steady-state. The sole presence of a non-induced iKMT2A-MLLT3 transgene did not significantly change the pattern indicating no impact on potential leakiness (Supplementary Fig. 1J). Analysis of the cell cycle profiles in specific subcompartments as defined by Evi1 expression revealed unexpected changes: while TPO stimulation resulted in a shift towards the G0 phase (+13.6%) of Evi1neg LT-HSC, Evi1high- and Evi1low LT-HSC showed

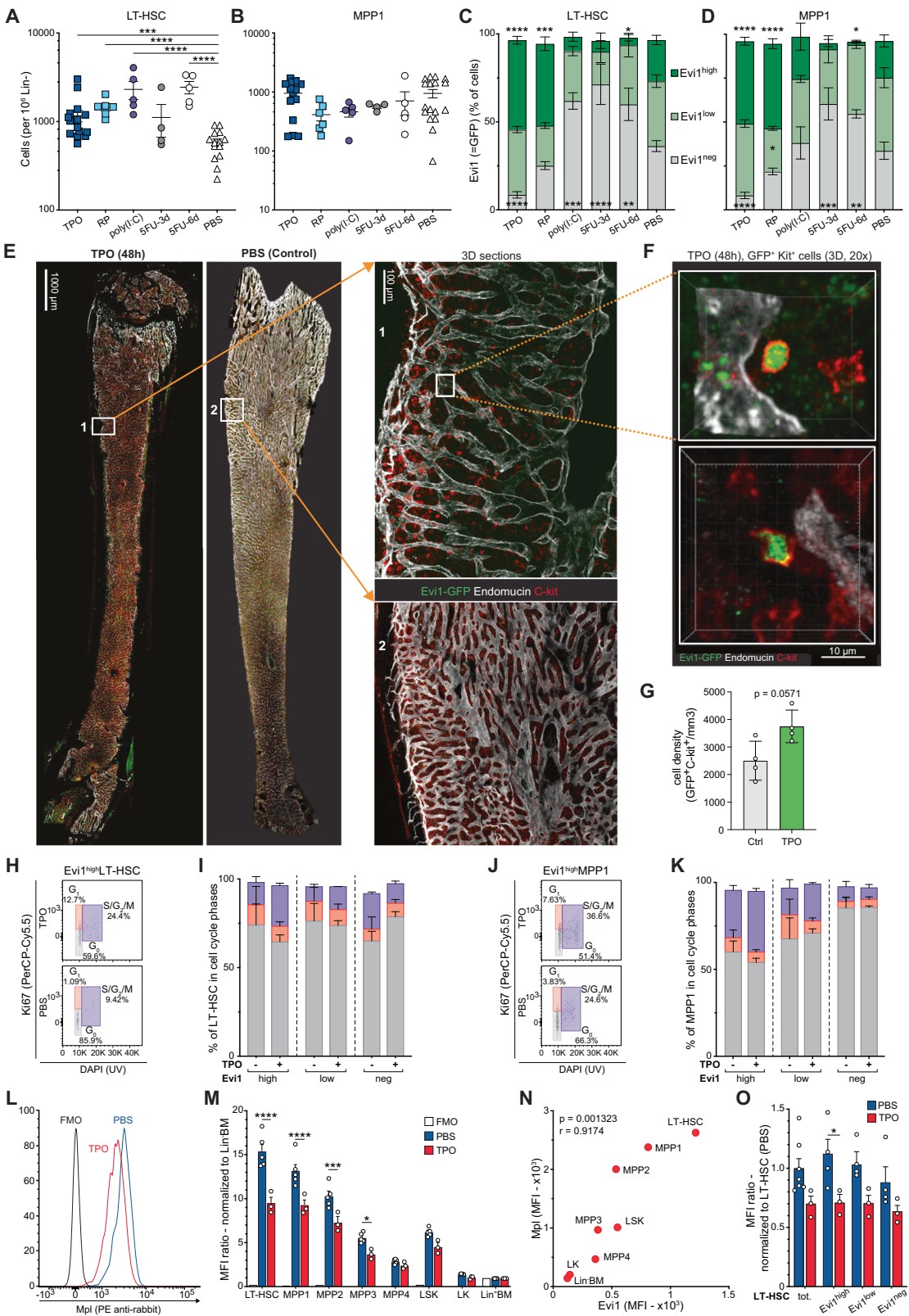

increased cycling (Evi1$^{high}$: +10.2%; Evi1$^{low}$: +4.8%) (Fig. 1H–I) with a similar trend observed for the Evi1$^{high}$ MPP1 cells (+7.7%) (Fig. 1J–K). These data suggest that a single injection of TPO selectively increases cycling of Evi1-expressing HSPC in vivo. In order to decipher the TPO activity on HSPC and more specifically on the Evi1$^+$ fraction, we stained Lin$^-$BM cells from Evi1-GFP mice for the TPO receptor (MPL)[27] (Fig. 1L). At steady-state, as expected[28], we observed significantly higher MPL

levels in the more quiescent immature populations (LT-HSC, MPP1, LT-HSC and MPP1: adj.pval<0.0001) and lower expression in the more mature cell populations (MPP2: adj.pval=0.0002, MPP3: adj.pval=0.0469) (Fig. 1M; Supplementary Fig. 1K). Notably, we found a strong correlation between the mean fluorescence intensity (MFI) of MPL and EI1 in the different HSPC subsets (Pearson coefficient $r = 0.9$, $p = 0.001$) (Fig. 1N). Injection of TPO 48 h prior to the analysis led to a

**Fig. 1 | TPO exposure increases cycling of Evi1$^{high}$ LT-HSC and MPP1 cell fractions.** Number of LT-HSC (**A**) and MPP1 (**B**) normalized to $10^6$ Lin⁻BM cells in animals treated with either TPO (dark blue, $n = 15$), RP (light blue, $n = 6$), poly(I:C) (lilac, $n = 5$), 5FU-3d (gray, $n = 4$), 5FU-6d (white dots, $n = 5$) or PBS (white triangles, $n = 14$ (**A**) and $n = 17$ (**B**). Flow cytometric quantification LT-HSC (**C**) and MPP1 (**D**) expressing Evi1$^{high}$, Evi1$^{low}$ and Evi1$^{neg}$ treated with TPO: $n = 15$, RP: $n = 6$, pl:pC: $n = 5$, 5-FU 3 d: $n = 4$, 5-FU 6 d: $n = 5$, PBS: $n = 13$ (**E**) Representative high-resolution image of femurs of KME mice after PBS (left) or TPO (right) injection on DOX for 5 days (starting 48 h after injection). Bones from 4 mice per condition stained with anti-endomucin (white), anti-cKit (red) and anti-GFP (green). Scale bar=1000 μm. **F** 3-D images of Evi1⁺(= GFP⁺) c-Kit⁺ cells. Cells of interest (Evi1⁺/Kit⁺) annotated based on nuclear expression of Evi1-GFP (green) and membranous expression of Kit (red). **G** Density of GFP⁺Kit⁺ cells in femurs from PBS- vs TPO-treated KME mice. Statistical significance determined using two-sided unpaired Mann–Withney test. (Ctrl, $n = 4$, TPO, $n = 4$). **H** Flow cytometry plots showing the fractions (%) of Evi1$^{high}$ LT-HSC in the $G_0$, $G_1$ and S/$G_2$/M phases 48 h after TPO or PBS treatment. **I** Flow cytometric quantification of the proportion (%) of LT-HSC in the $G_0$, $G_1$ and S/$G_2$/M phases depending on Evi1 expression levels (high, low or negative) 48 h after PBS ($n = 2$) or TPO ($n = 3$) treatment. **J** Flow cytometry plots showing the fractions (%) of Evi1$^{high}$ MPP1 in the $G_0$, $G_1$ and S/$G_2$/M phases 48 h after TPO or PBS treatment. **K** Flow cytometric quantification of the proportion (%) of MPP1 in the $G_0$, $G_1$ and S/$G_2$/M phases depending on Evi1 expression levels (high, low or negative) 48 h after PBS ($n = 2$) or TPO ($n = 3$) treatment. **L** Representative histogram showing the mean fluorescence intensity (MFI) of Mpl surface expression on LT-HSC in the different tested conditions (FMO = Fluorescence Minus One). **M** Flow cytometric quantification of Mpl expression in HSPC of PBS ($n = 5$), and TPO ($n = 3$)- treated animals. Values normalized to Lin⁻ BM MFI (FMO = Fluorescence Minus One). **N** Correlation between Mpl and Evi1 expression in HSPC populations ($n = 7$) based on MFI. Pearson correlation: $r = 0.9174$, $p = 0.001323$ (two-sided). **O** Mpl expression (MFI) in LT-HSC expressing different levels of Evi1 from TPO-treated mice normalized to LT-HSC from PBS-treated mice. (Evi1$^{high}$ LT-HSC/LT-HSC MFI ratio; PBS: $1.12 \pm 0.12$, $n = 7/4$; TPO: $0.71 \pm 0.07$, $n = 3/3$, $p = 0.04$). $N$ = number of mice. Statistically significant differences between treatments and PBS were calculated by 1-way ANOVA (**A**–**B**) or 2-way ANOVA (**C**–**D**, **I**, **K**, **M**, **O**) followed by Sidak multiple comparison test (**M**, **O**), Tukey's post-test (**C**–**D**) and two-sided unpaired $t$-test (**A**–**B**, **I**, **K**) and data are presented as mean ± SEM (*$p < 0.05$; **$p < 0.01$; ***$p < 0.001$; ****$p < 0.0001$). Source data provided in the Source Data file.

---

significant reduction of MPL surface expression mostly confined to the Evi1$^{high}$ population, in particular the Evi1$^{high}$ LT-HSC (Fig. 1O). A decrease of MPL expression was also detected in all other EI1-expressing HSPC populations (Supplementary Fig. 1L–N) reflecting classic autoregulation[29]. Together, our data indicates that TPO selectively increases the fraction of Evi1$^{high}$ HSC by increasing cycling of cells expressing the highest MPL levels.

## TPO increases colony formation and accelerates iKMT2A-MLLT3-driven AML

To determine the impact of in vivo TPO stimulation on iKMT2A-MLLT3 transformation, we compared colony formation by BM cells isolated from mice 48 h after TPO or PBS injection. TPO-exposed cells formed significantly more colonies with often a previously described invasive "type IV" phenotype formed by cells with an immature morphology (Evi1$^{high}$ LT-HSC: $69 \pm 5.9$ vs $29.4 \pm 7.5$, adj.p < 0.0001; Evi1$^{high}$ MPP1: $73.17 \pm 4.8$ vs $38.75 \pm 9.8$, adj.p < 0.0001). (Fig. 2 A, B; Supplementary Fig. 2 A)[9]. After 1 week in methylcellulose cultures (MC), colony-forming cells from TPO-treated donors maintained similar Evi1 levels as Evi1$^{high}$ cells from control donors (Fig. 2C). Overall, induction of iKMT2A-MLLT3 significantly increased colony formation by in vivo TPO-stimulated LT-HSC and MPP1 (adj.$p = 0.0451$).

To address the impact of exogenous TPO on in vivo induction of iKMT2A-MLLT3-driven AML, we harvested BM cells 48 h after a single TPO injection and transplanted equal numbers ($10^3$) of naïve (CD45.2⁺) KME LT-HSC or MPP1 into lethally irradiated (CD45.1⁺) wildtype recipients (Fig. 2D). It is of note that due to the almost selective expansion of Evi1$^{high}$ cells by TPO (Fig. 1C–D), mice transplanted with TPO-stimulated HSC obtained a very similar number of Evi1$^{high}$ cells than those transplanted with flow-enriched Evi1$^{high}$ cells. Recipients of TPO-stimulated LT-HSC or MPP1 developed disease symptoms significantly earlier than mice that received Evi1$^{high}$ cells from PBS-treated mice (Fig. 2E). Diseased animals exhibited the signs of iKMT2A-MLLT3 AML with reduced body weight, pronounced splenomegaly and multi-organ infiltration of leukemic blasts (Fig. 2F–G; Supplementary Fig. 2B–H). While no differential Mac-1/Gr-1 or c-Kit expression was detected, faster disease development in the recipients of TPO-exposed MPP1 was associated with an increased proportion of FcγRII/III⁻ cells ($1.7\% \pm 0.5$ vs $6.7\% \pm 1.2$, $p = 0.0076$) (Supplementary Fig. 2I–L) with tumor cells expressing high levels of Evi1 in most animals (Fig. 2H). Comparison of the Evi1⁺ blasts in symptomatic recipients derived from HSC with or without TPO exposure (Fig. 2I) showed that, in almost half (18 out of 39) of the recipients of TPO-stimulated cells, over 25% of the blasts expressed Evi1. By contrast, only a quarter of the mice (7 of 29) transplanted with flow-sorted Evi1$^{high}$ cells from PBS-treated donors

showed Evi1 expression above that threshold. Evi1 expression significantly correlated with disease latency (Fig. 2J–K) in the recipients of TPO-stimulated cells (+TPO: $r = -0.45$, $p = 0.007$; -TPO: $r = -0.51$, $p = 0.05$). Overall, we found that exposure to exogenous TPO accelerates HSC-derived iKMT2A-MLLT3-driven AML.

## TPO stimulation rapidly modulates gene expression of iKMT2A-MLLT3 HSPC

To investigate the immediate impact of in vivo exposure to exogenous TPO on gene expression, we profiled Evi1$^{high}$ Flt3⁻ LSK iKMT2A-MLLT3 cells after 2 and 5 days on doxycycline (DOX) at the single cell level (Fig. 3A). We focused on LT-HSC and MPP1-3 by using a hashtag oligonucleotide and antibody-derived tag labeling strategy for cellular indexing of transcriptomes and epitopes by sequencing (CITE-seq). We sequenced 71'261 and 32'198 cells from day 2 and day 5, respectively. Integrative bioinformatic analysis defined 20 cell clusters which have been annotated using previously published mouse haematopoiesis signatures[30–34] (Fig. 3B–C; Supplementary Fig. 3A) indicating that the HSPC expression profiles form more a continuum cloud rather than clearly separated transcriptionally defined cell clusters. Following the strategy by Fast et al.[34], we distinguished primitive ("LT-HSC", "ST-HSC_1" and "ST-HSC_2") and more mature myeloid cell clusters (e.g., GMP_1, GMP_2, and others) (Fig. 3B). Additionally, we identified specific clusters ("MPP2_2", "MPP2_3", "MPP3_4" and "CMP_1") containing a significant fraction of cycling cells, mostly in $G_2$/M phase (Fig. 3B; Supplementary Fig. 3A). The expression analysis for HSPC after two days of iKMT2A-MLLT3 expression resulted in few differentially expressed genes (FDR < 0.05) in 7 clusters of TPO- compared to PBS-treated-cells (Fig. 3D). Notably, the major histocompatibility complex (MHC) class II -associated molecule Cd74 and the suppressor of cytokine signaling 2 (Socs2) both higher expressed in cluster "ST-HSC_2" were previously connected to AML aggressiveness and stemness[35,36] (Fig. 3E, Supplementary Data files 1 and 2). MPP1_2 cells were characterized by higher expression of Cd52 which has been previously proposed as immunotherapy target in EVI1$^{high}$ AML[37] (Fig. 3F, Supplementary Data 1 and 2). Gene set analysis (GSEA) revealed up-regulation of HOXA9- and Interferon/STAT-related pathways (Fig. 3G; Supplementary Data File 3). Also, 5 days of iKMT2A-MLLT3 activation resulted in differential expression of a limited number of genes in 5 clusters (Fig. 3H). In the primitive "LT-HSC" stem cell cluster we observed higher expression of the homeobox transcription factor Pbx3 associated with leukemic stemness, and lower expression of the arginine methyltransferase Prmt6, a proposed negative regulator of Hox and Myc[38,39] (Fig. 3I, Supplementary Data File 4). In addition, the

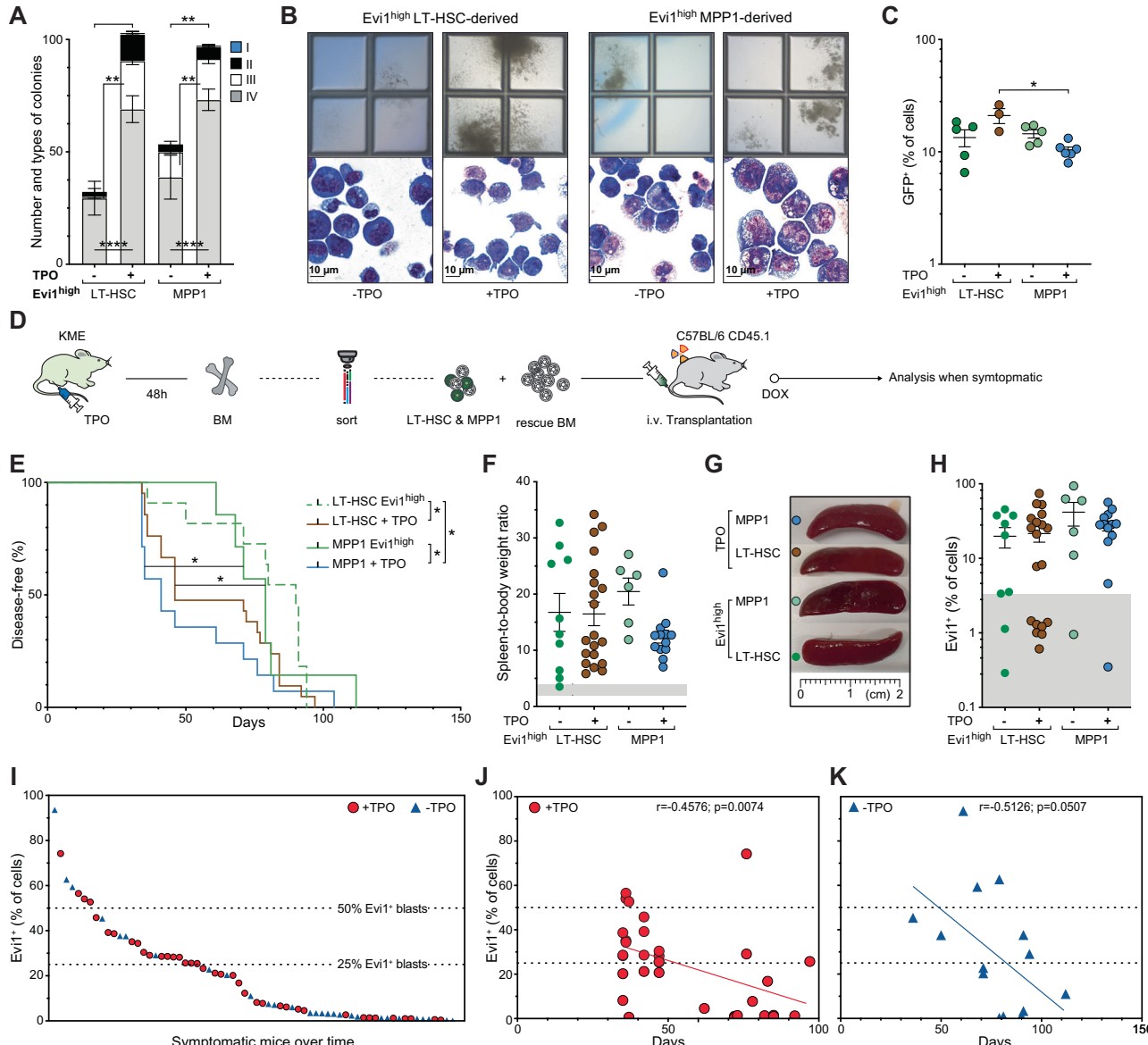

**Fig. 2 | TPO increases colony formation and accelerates iKMT2A-MLLT3-driven AML. A** Number and types of colonies formed by Evi1^high LT-HSC and Evi1^high MPP1 from KME mice (on DOX) harvested 2 days after TPO (200μg/kg BW) or PBS treatment (LT-HSC (PBS): *n* = 5, LT-HSC (TPO): *n* = 4, MPP1 (PBS): *n* = 4, MPP1 (TPO): *n* = 6). More "type IV" colonies were formed by TPO-exposed cells. *n*= number of MC. **B** Representative pictures of colonies (2.5x) (top) and images of cytospin preparations (60x) (bottom) of Evi1^high LT-HSC- (left) and Evi1^high MPP1-derived-cells (right) from mice with or without TPO treatment grown for 10 days. Scale bar = 10 μm. **C** Proportion (%) of Evi1^+ ( = GFP^+) cells in Evi1^high LT-HSC (green & brown)- and Evi1^high MPP1 (light green & blue)-derived colony-forming cells from mice with or without TPO treatment. LT-HSC: TPO (*n* = 3), PBS (*n* = 5); MPP1: TPO (*n* = 6), PBS (*n* = 5). **D** Experimental outline: transplantation of sorted LT-HSC or MPP1 from TPO-stimulated KME mice wildtype recipients (**E**) Kaplan–Meyer plot of disease-free mice. Transplantation of sorted HSCP from TPO-treated mice resulted in accelerated disease development: MPP1: 41 d vs 79 d, *n* = 14 vs *n* = 7, *p* = 0.025; LT-HSC: 46 d vs 90 d, *n* = 21 vs *n* = 11, *p* = 0.028; Mantel-Cox test. **F** Spleen-to-BW ratio

and (**G**) representative pictures of spleens from diseased mice transplanted with Evi1^high LT-HSC or -MPP1 with and without TPO pre-treatment: LT-HSC-derived: TPO (*n* = 20), PBS (*n* = 10); MPP1-derived: TPO (*n* = 13), PBS (*n* = 6). Gray area in (**F**): spleen size in healthy mice. **H** Proportion (%) of Evi1^+ ( = GFP^+) BM cells from diseased mice transplanted with LT-HSC or MPP1 from TPO-treated or untreated controls. LT-HSC-derived: TPO (*n* = 19), PBS (*n* = 9); MPP1-derived: TPO (*n* = 13), PBS (*n* = 6). (**I**) Flow cytometric quantification of the proportion (%) of Evi1^+ leukemic cells in diseased animals transplanted with LT-HSC or MPP1 from TPO- or PBS-treated donors. **J–K** Correlation between disease latency and the proportion of Evi1^+ BM cells in diseased mice transplanted with (*n* = 19) (**J**) or without (*n* = 23) TPO-treated (**K**) cells. Two-sided Pearson correlation test was used to calculate the significance. *N* = number of mice. Statistical significance was calculated with 1-way ANOVA (**C**, **F**, **H**) and 2-way ANOVA (**A**) followed by Tukey's post-hoc were used to test for significance and data are represented as mean ± SEM (*\*p* < 0.05; *\*\*p* < 0.01; *\*\*\*p* < 0.001; *\*\*\*\*p* < 0.0001). Source data are provided in the Source Data file.

cycling clusters "MPP2_3" and "MPP3_4" where characterized by upregulation of the transcription factors Fos (only in cluster "MPP2_3") and Junb (Fig. 3J, Supplementary Data File 4) both activated by TPO/MpPL signaling and involved in cell invasion, and often overexpressed in AML[40,41]. Very similar to day 2, GSEA showed upregulation of genes in pathways associated with Hoxa9

("TAKEDA_Targets of NUP98-HOXA9_fusion_UP"), $G_2$/M checkpoints and Ifnα and Myc activation (HALLMARK pathways) (Fig. 3K; Supplementary Data File 3). Altogether, these data illustrate that a single TPO injection rapidly modulates the gene expression program of DOX-induced iKMT2A-MLLT3 HSC leading to dysregulation of genes downstream of Tpo/Mpl and the fusion including Hoxa9.

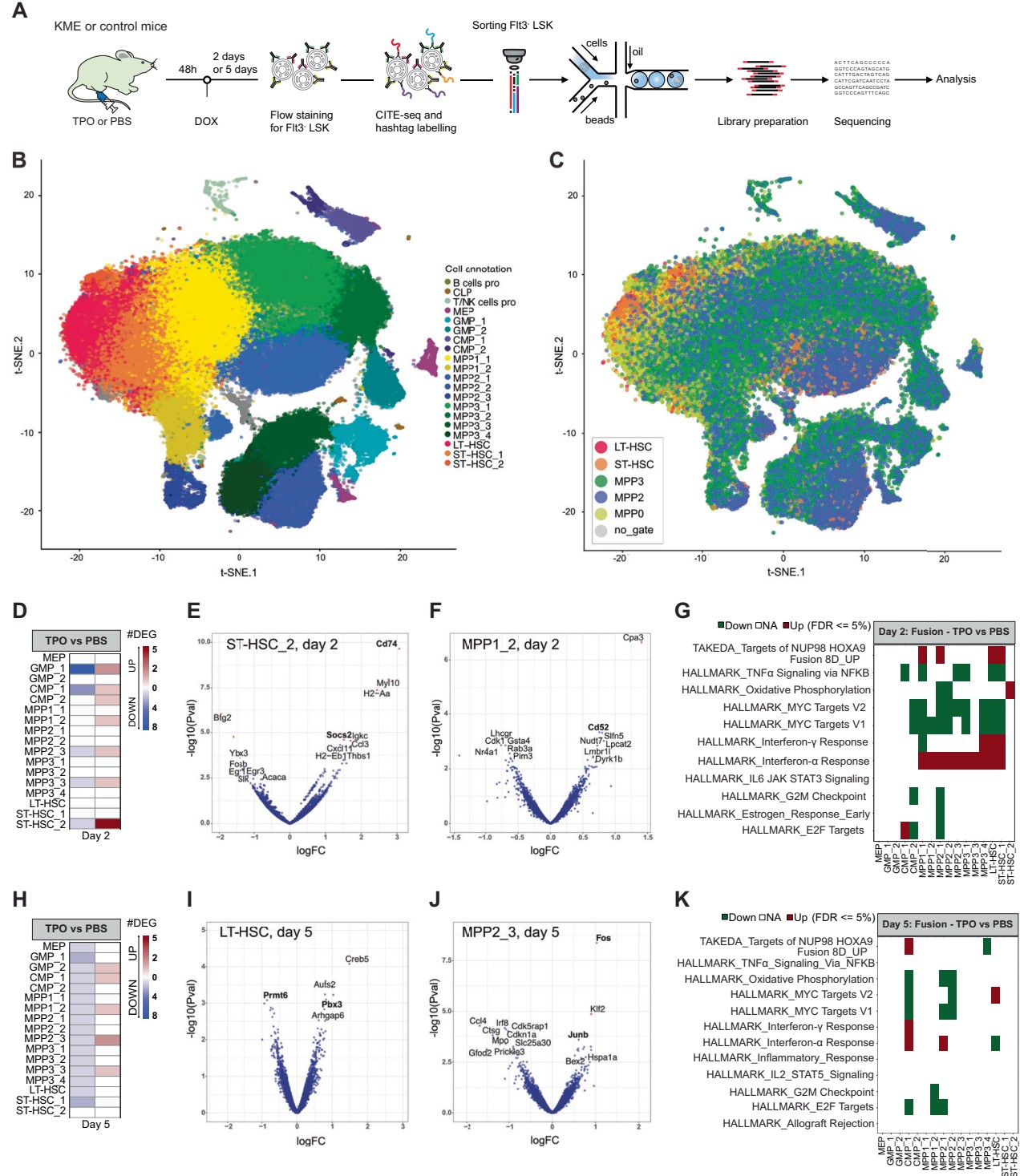

**Fig. 3 | TPO stimulation rapidly modulates gene expression of iKMT2A-MLLT3 HSPC. A** Experimental setup: Flt3⁻LSK cells were isolated 48 h after a single dose of TPO followed by 2 or 5 days on DOX to induce expression of the iKMT2A-MLLT3 fusion. RNA from Flt3⁻ LSK cells was sequenced at the single cell level using hash-tagged oligos (HTO) and CITE-seq antibodies (ADT). **B** T-SNE dimensionality reduction plot, with color illustrating the annotation of the iKMT2A-MLLT3 HSPC clusters based on published transcriptomic signatures[30,89,90]. **C** T-SNE dimensionality reduction plot, with color illustrating the gating of the cells using CITE-seq markers, following the strategy of Fast et al.[39]. **D** Heatmap showing the number of down-regulated (left column) and up-regulated (right column) DEGs (FDR < 0.05) in TPO- vs PBS exposed cells after 2 days on DOX. Volcano plots of DEGs in TPO- vs PBS exposed ST-HSC_2 (**E**) or MPP1_2 (**F**) cell clusters after 2 days on DOX.

Significant DEG (FDR < 0.05) are colored in red. *P*-values were calculated with likelihood ratio tests from EdgeR package. **G** MSigDB pathway analysis (FDR < 0.05) of DEGs in TPO- vs PBS-exposed HSPC after 2 days on DOX. *p*-values were calculated with two-sided *t*-test of the camera function from the limma R package. **H** Heatmap showing the number of down-regulated (left column) and up-regulated (right column) DEGs (FDR < 0.05) in TPO- vs PBS exposed HSPC after 5 days on DOX. Volcano plots of DEGs in TPO- vs PBS-exposed LT-HSC (**I**) or MPP2-3 (**J**) after 5 days on DOX. Significant DEG (FDR < 0.05) are colored in red. *p*-values were calculated with likelihood ratio tests from EdgeR package. **K** MSigDB pathway analysis (FDR < 0.05) of DEGs in TPO- vs PBS-exposed HSPC after 5 days on DOX. *p*-values were calculated with two-sided *t*-test of the camera function from the limma R package.

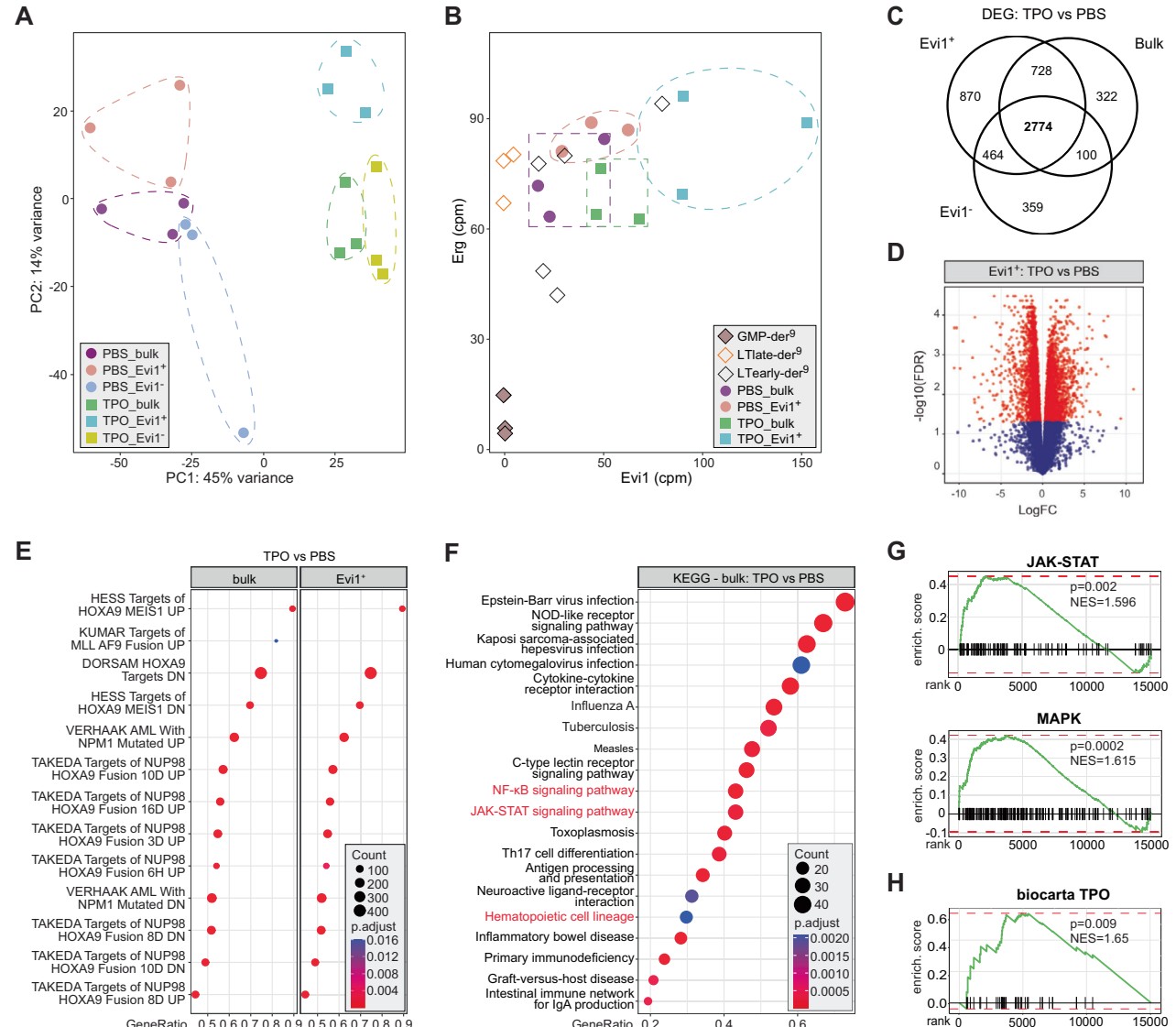

**Fig. 4 | The iKMT2A-MLLT3 AML expression signature reflects the origin of TPO-exposed HSC. A** PCA shows a marked separation of the gene expression signatures of KME AML cells by TPO exposure of the donors with only a minor influence of Evi1 expression. **B** AML cells emerging from TPO-treated KME BM expressed similar or higher Evi1 and Erg levels than LT-HSC, or GMP-derived iKMT2A-MLLT3 blasts reported earlier[9]. **C** Venn diagram showing the number of common DEGs the different AML cell fractions (Evi1+, Evi1-, and bulk) emerging from TPO- vs. PBS-stimulated KME BM cells. **D** Volcano plot of DEGs in Evi1+ AML cells emerging from TPO- or PBS-stimulated KME BM cells. **E** GSEA (MSigDB) analysis of DEGs in AML cells emerging from TPO- or PBS-treated KME BM cells

revealed associations to signatures of aberrant Hoxa9 activity in both, the bulk and Evi1+ cell fractions. **F** KEGG analysis of DEGs in AML cells (bulk) emerging from TPO- or PBS-treated KME BM cells revealed up-regulation JAK-STAT and NF-κB signaling pathways. **G** GSEA revealed significant upregulation of JAK-STAT (top) and MAPK (bottom) pathways related genes in bulk blasts from symptomatic mice transplanted with TPO-treated KME BM cells. **H** GSEA revealed significant upregulation of the Biocarta (MSig-DB) TPO pathway-related genes in bulk blasts from symptomatic mice transplanted with TPO-treated KME BM cells. p-values were calculated with quasi-likelihood F-tests from EdgeR package (**D**) or permutation test of GSEA function from clusterProfiler R package (**E–H**).

## The iKMT2A-MLLT3 AML expression signature reflects the origin of TPO-exposed HSC

To characterize the impact of exogenous TPO on HSC-derived iKMT2A-MLLT3-driven AML, we compared the gene expression signatures from emerging leukemic cells ("bulk") (Supplementary Fig. 4A). Principal component analysis (PCA) showed a clear separation of AML emerging from TPO- vs PBS-exposed HSPC (PC1: 45%) (Fig. 4A). Evi1+ cells clustered apart from Evi1- or bulk cells. We previously showed that mRNA expression levels of Mecom and its putative target Erg define the cellular origin of iKMT2A-MLLT3 AML[9,42]. Notably, AML cells derived from TPO-exposed HSC also expressed increased levels of Evi1 and Erg that reached similar or even higher levels than previously seen in LT-HSC-derived iKMT2A-MLLT3 AML[9] (Fig. 4B).

Exposure to exogenous TPO resulted in 2774 differentially expressed genes (DEG) (FDR < 0.05) independent of Evi1 expression (Fig. 4C; Supplementary Data Files 5–7). Evi1+AML cells derived cells from TPO-exposed HSC compared to cells from PBS-treated animals revealed 4836 DEG (2187 higher and 2649 lower expressed; FDR < 0.05) (Fig. 4D; Supplementary Fig. 4B–D). GSEA indicated significant correlations to signatures from haematopoietic cells transformed by overexpression of MEIS1/HOXA9 or KMT2A-MLLT1, as well as cells carrying mutant NPM1 or the NUP98-HOXA9 fusion gene all converging at activation of the HOXA gene cluster[43] (Fig. 4E). The DEG signatures of iKMT2A-MLLT3 AML cells derived from TPO-exposed HSC (bulk and GFP+ fraction) were characterized by higher expression of multiple genes downstream of the TPO receptor MPL. KEGG pathway enrichment

analysis showed significant enrichment of the "JAK-STAT", "MAPK" and "NF-κB" signaling pathways (Fig. 4F–G; Supplementary Fig. 4E–F). The significant association to TPO/MPL-mediated transcriptional programs was substantiated by GSEA taken from Biocarta (MSig-DB) and a custom gene list based on a published TPO signaling map[44] (Fig. 4H; Supplementary Fig. 4G–H). These observations suggest that TPO significantly impacts the gene expression program of HSC-derived iKMT2A-MLLT3-driven AML cells.

## Comparing expression signatures from TPO-stimulated HSC-derived murine Evi1+AML with human EVI1+ AML reveals common HSC genes associated with poor outcome

To explore whether the gene expression signature of Evi1+ iKMT2A-MLLT3 AML derived from TPO-stimulated HSC is reflected in human EVI1+ AML, we interrogated four public gene expression databases including the TARGET, BEAT, LEUCEGENE and the recently published patient cohorts from the ST.JUDE hospital[45–48]. Measurable (log2(RPKM) > 0) MECOM expression was found in 1164 patients of which 132 contained rearrangements of the KMT2A- and 24 of the MECOM gene loci respectively (Supplementary Data file 8). Arranging all selected patients according to expression of EVI1 and ERG mRNA showed very similarly to the iKMT2A-MLLT3 mouse model (Fig. 4B), a continuum from EVI1low to EVI1high AML. We therefore defined EVI1high-, EVI1intermediate- and EVI1low AML patients by k-means clustering and established the optimal number of clusters for each patient-derived dataset (Fig. 5A; Supplementary Fig. 5A).

We then determined DEG between EVI1high and EVI1intermediate patient clusters (FDR < 5%) and compared them with DEG from Evi1+ iKMT2A-MLLT3 murine AML originating from TPO- vs PBS-stimulated HSC. We found 1328, 9647, 1444 and 4075 DEG for the patients from the TARGET, BEAT, LEUCEGENE and ST.JUDE cohorts, respectively. Amongst them, 54, 255, 131 and 5533 genes were similarly dysregulated in the murine signatures (Fig. 5B; Supplementary Fig. 5B). Out of the common DEG, 47 genes were significantly higher expressed in the iKMT2A-MLLT3 mouse model and in at least two AML patient cohorts. Several of these DEG have been implicated in leukemic stemness, inflammation and chemoresistance. Three genes, MECOM, IL12Rβ2 (interleukin 12 receptor subunit beta 2), INPP4B (inositol polyphosphate-4-phosphatase type II B) were significantly higher expressed in mouse Evi1+ iKMT2A-MLLT3 AML and all human EVI1+ AML datasets (Fig. 5C, Supplementary Data file 9). Notably, although primarily known for its role as immune cell regulators, IL12 has been previously been proposed to play a role in HSC maintenance and self-renewal[49,50]. INPP4B has been proposed as potential EVI1 target and implicated in regulation of AML growth, proliferation and chemoresistance related with poor outcome[51,52]. Correlation analysis on the different cohorts showed that patients with high expression of MECOM also expressed high levels of the aforementioned genes (Fig. 5D–E; Supplementary Fig. 5C). However, although we found a significant correlation between MECOM and IL12RB2 expression (R > 0.5, p > 0.2) for patients with KMT2A-r and KMT2A-MLLT3 alterations in all AML databases, we did not see any correlation between INPP4B and MECOM expression in adult patients with KMT2A-MLLT3 (BEAT and LEUCEGENE cohorts).

Among the DEG from the mouse model that were significantly higher expressed in at least 3 human AML cohorts were ADGRG6 (adhesion G protein-coupled receptor G6, aka GPR126), PBX1 (PBX homeobox 1) and STYK1 (Serine/Threonine/Tyrosine kinase 1) (Supplementary Data 9). Similarly to INPP4B, ADGRG6 was reported to be expressed in leukemic stem cells in AML patients, and was found to be oncogenic in different models related to invasion and metastasis[53]. Notably, all these genes are also expressed at highest levels in HSC in normal haematopoiesis (Bloodspot) which most likely reflects the origin of the disease. In addition to common higher expressed DEG, we also defined 50 genes that were lowest expressed in the mouse model

and ≥2 AML patient cohorts. Some genes like FNDC3B (Fibronectin Type III Domain Containing 3B), MS4A3 (Membrane Spanning 4-Domains A3), P2RY2 (Purinergic Receptor P2Y2) were lowest expressed in normal HSC (Supplementary Data file 10). Separation of the patients based on median expression of the selected genes showed a significant survival disadvantage in paediatric and adult patients with higher expression of up-regulated DEG, particularly EVI1, IL12Rβ2 and INPP4B (Fig. 5F; Supplementary Fig. 5D).

GSEA and over-representation analysis (ORA) on the common DEG between human and mouse AML revealed enrichment of pathways linked to several "AML" signatures, "Haematopoietic stem cells" and targets of HOXA9 ("TAKEDA Targets of NUP98-HOXA9 fusion") (Supplementary Fig. 5E). Specifically, we found an upregulation of signatures of "HOXA9-MLL/ENL", "Targets of MLL-AF9", "NUP98-HOXA9 fusion" and "HOXA9-binding sites by ChIP-seq" (Supplementary Fig. 5F). Collectively, comparative cross-species expression profiling revealed several HSC genes that characterize human EVI1+ AML with poor outcome.

## Functional validation of common DEG in human KMT2A-r AML cells

To functionally evaluate the impact of common DEG in EVI1+ iKMT2A-r and human AML we identified a single KMT2A-rearranged (KMT2A-MLLT1+) cell line OCI-AML4 that expresses high levels of EVI1 without any 3q26 locus alteration (https://depmap.org/portal/ & Fig. 6A)[54]. In contrast to MECOM-rearranged human AML lines, OCI-AML4 cells express significant levels of MPL. Although OCI-AML4 cells grow in the absence of TPO, short-term TPO stimulation of starved cells results in activation of the STAT pathway as seen by increased phosphorylation of STAT5. In addition, TPO stimulation was accompanied with an increase of nuclear HOXA9 confirming earlier observations[55] (Supplementary Fig. 6A–C).

To explore a potential dependency of OCI-AML4 cells on origin-related genes, we expressed inducible shRNA coupled to mCherry expression, selectively targeting MECOM, MPL, HOXA9, as well as IL12Rβ2 and INPP4B. The human KMT2A-MLLT3-rearranged cell line MOLM-13 lacking MECOM and MPL expression served as negative control (Fig. 6A). KD of MECOM significantly impaired the growth of OCI-AML4 in liquid culture, without significantly impacting mRNA expression of the other genes (Fig. 6B–C). MPL KD slightly impaired expansion of OCI-AML4. Notably, we did not observe any effects on MPL and MECOM mRNA expression that would indicate cross-regulation (Fig. 6B). OCI-AML4 cells were particularly dependent on MECOM, with significant selection against the KD, as seen by the reduction in mCherry-expressing cells (Supplementary Fig. 6I) but also by the reduced overall number of colonies (Fig. 6D). Both cell lines express high levels of IL12Rβ2 (https://depmap.org/portal/) but are not dependent on the presence of TPO or IL12 for growth. Interestingly, KD of IL12Rβ2 did impair growth in liquid culture (Fig. 6C), and even more colony formation by OCI-AML4 cells in MC (Fig. 6D–E). IL12Rβ2 appears to be essential for OCI-AML4, as negative selection was observed with the significant reduction in mCherry-expressing cells (Supplementary Fig. 6I). Notably, IL12Rβ2 KD resulted in a ˜50% decrease in MECOM mRNA expression (Fig. 6B), suggesting potential cross-regulation. shRNA-mediated targeting of INPP4B reduced growth of OCI-AML4 and MOLM-13 cells (Fig. 6C, F), and reduced colony formation by both of them without reaching significance (Fig. 6D–G). We did not observe any significant reduction of growth by HOXA9 KD of OCI-AML4 and MOLM-13 cells. Notably, we observed a slight increase of colonies associated with decreased expression of MPL in OCI-AML4 cells (Fig. 6B–E).

As the IL12Rβ2 KD effects in AML cells are unexpected we further analysed it in KMT2A-MLLT3+ THP-1, and HL-60 AML cells that lack any KMT2A-rearrangement. Notably, KD of IL12Rβ2 did not significantly alter growth in liquid cultures but reduced colony formation in MC by

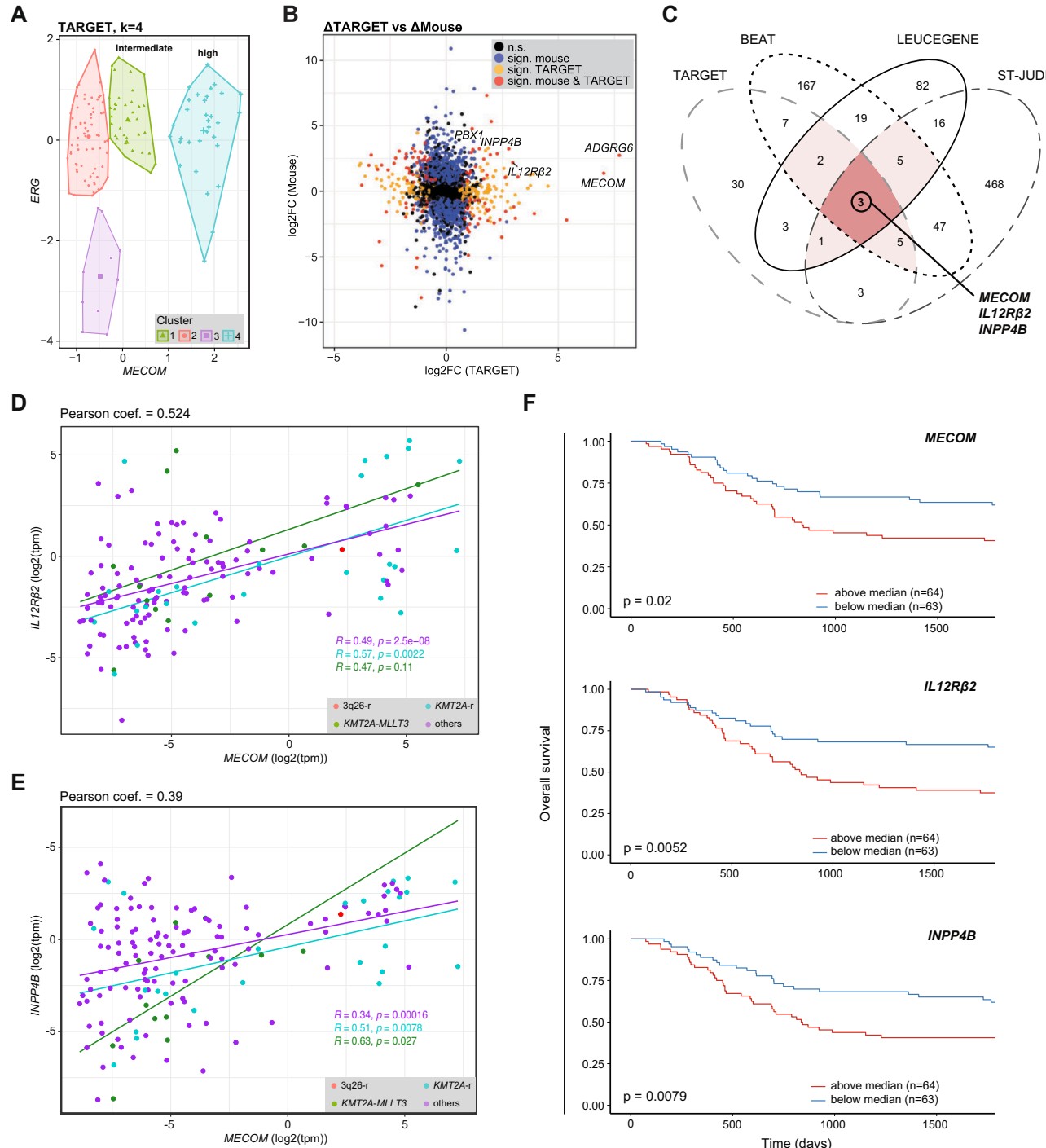

**Fig. 5 | Comparison of expression signatures from TPO-stimulated HSC-derived murine Evi1⁺AML with human EVI1⁺AML reveals common HSC genes associated with poor outcome. A** K-means clustering according to ERG and MECOM expression levels of the AML patients from the TARGET cohort[46]. **B** Scatterplot comparing DEGs of TPO- vs PBS-exposition derived iKMT2A-MLLT3 murine AML with DEGs of EVI1^high vs EVI1^intermediate patients from the TARGET cohort[46]. **C** Venn diagram illustrating similarities between DEGs in the human EVI1⁺AML and the mouse Evi1⁺AML signatures from 4 AML patient cohorts (TARGET, BEAT, LEUCE-GENE, ST.JUDE)[45–48]. Scatterplots showing correlations of selected top up-regulated genes IL12Rβ2 (**D**) and INPP4B (**E**) compared to MECOM in the AML patients of the TARGET cohort. Pearson correlations were calculated for all patients (on the top), or only in group of patients with KMT2A-r, in KMT2A-MLLT3 or without KMT2A genomic rearrangements (colored in blue, green and purple respectively). Two-sided Pearson correlation test was used to calculate the significance for each group. **F** Survival curves for AML patients from the TARGET cohort split according to the median gene expression (TPM) of MECOM (top), IL12Rβ2 (middle) and INPP4B (bottom). p-values were calculated with Log-rank test.

both cell lines, while KD of MECOM had no significant effects as expected (Fig. 6I–N).

Further validation of these dependencies in primary Evi1^high iKMT2A-MLLT3 AML cells from diseased mice turned out to be technically challenging. Nevertheless, we found that inactivation of Mecom

and Il12rβ2 both resulted in reduced clonogenic growth in MC with a shift from type I to more type II colony formation (Fig. 6O–R). Collectively, these validation studies indicate a significant dependency on MECOM of EVI1^high KMT2A-r AML cells while IL12Rβ2-dependency seems EVI1-independent.

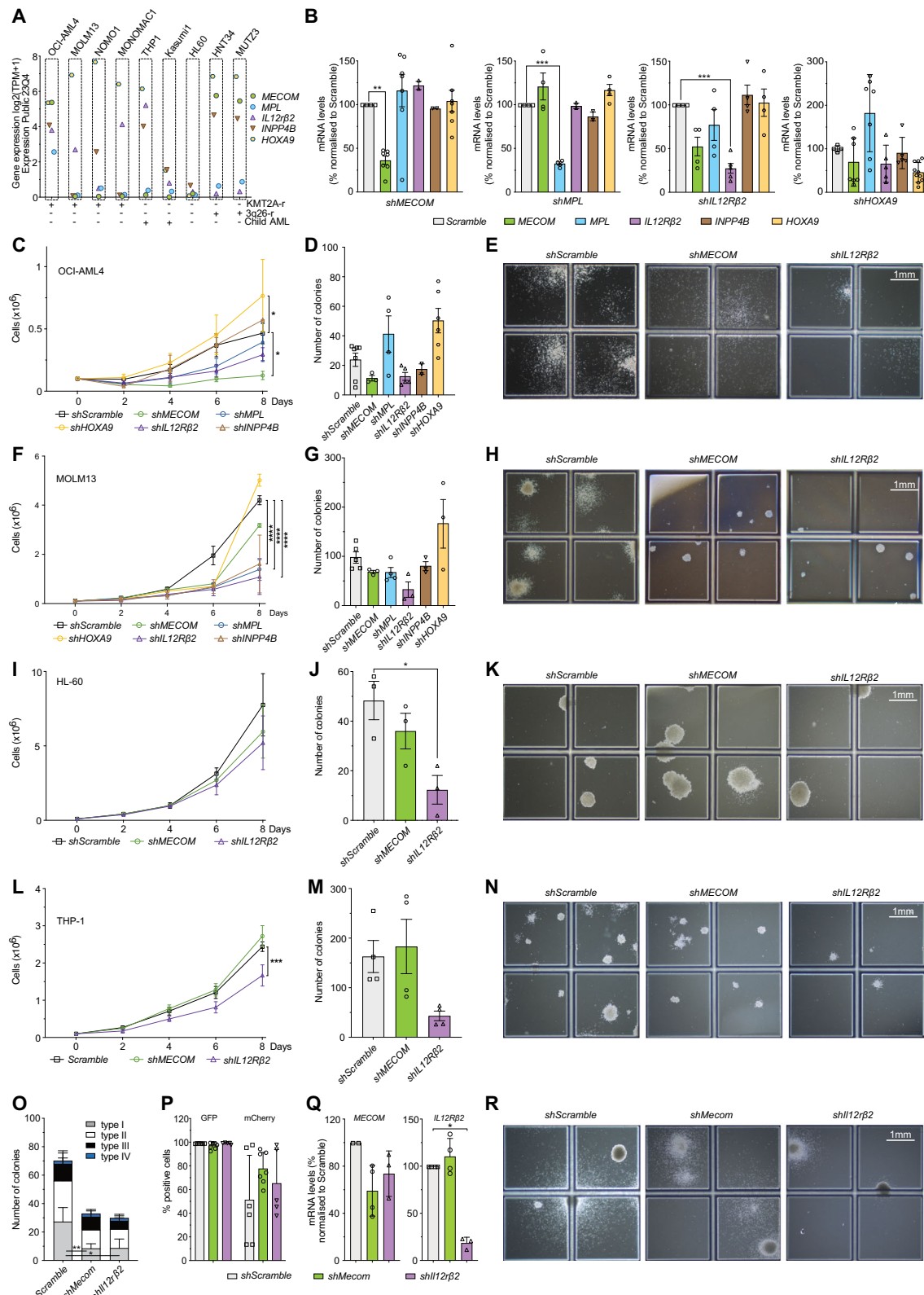

## Discussion

Integration of an Evi1-GFP reporter allele in the inducible iKMT2A-MLLT3 mouse line allowed us to demonstrate that a single dose of the cytokine TPO significantly and specifically increased the fraction of cycling Evi1high HSC and accelerated disease induction of iKMT2A-MLLT3 AML.

Earlier work indicated that KMT2A fusion oncogenes primarily exert their transforming potential by inducing aberrant self-renewal in myeloid progenitor cells. Inherently faster cell cycle progression was proposed to make GMP more permissive for transformation by retrovirally expressed rKMT2A-MLLT3 than more quiescent cells higher up in the hematopoietic hierarchy[4]. These effects were most likely

**Fig. 6 | Functional validation of top-upregulated DEG in human and mouse KMT2A-r EVI1⁺AML cells. A** Expression of selected common higher expressed DEGs in mouse and human Evi1/EVI1^high AML (MECOM, MPL, IL12Rβ2, INPP4B and HOXA9) in different human AML cells (DepMap 23Q4 Public). Note: OCI-AML4 is the only KMT2A-r cell line expressing significant levels of all genes. **B** RT-qPCR assessed mRNA levels of MECOM, MPL, IL12Rβ2, INPP4B and HOXA9 in OCI-AML4 virally expressing shRNA targeting MECOM, MPL, IL12Rβ2, HOXA9 and scramble control. The values represent the average of at least 2 shRNA/target (Supplementary Fig. 6E−H). Growth curves of OCI-AML4 (**C**), MOLM-13 (**F**), HL-60 (**I**), THP-1 (**L**) AML cells expressing the indicated shRNA. Only the statistically significant differences (based on adjusted *p*-value from Dunnett's test) are marked. Number of colonies formed by OCI-AML4 (**D**), MOLM-13 (**G**), HL-60 (**J**), THP-1 (**M**) AML cells expressing the indicated shRNA 14 days after plating in MC (H4534). Representative pictures of colonies formed by OCI-AML4 (**E**), MOLM-13 (**H**), HL-60 (**K**), THP-1 (**N**) cells expressing the indicated shRNAs after 14 days in MC (2.5x magnification). **O**

Number and types of colonies formed by Evi1^high iKMT2A-MLLT3 AML cells emerging from TPO-stimulated BM expressing the indicated shRNA, after 7 days in MC (M3534). **P** Percentage of GFP- or mCherry-positive Evi1^high iKMT2A-MLLT3 AML cells expressing Mecom, Il12rβ2 or Scramble shRNA harvested from MC plates. **Q** Mecom and Il12rβ2 mRNA expression in Evi1^high iKMT2A-MLLT3 AML cells expressing the respective shRNA harvested from MC plates. **R** Representative pictures from colonies formed by Evi1^high iKMT2A-MLLT3 AML cells expressing the different shRNAs after 7 days in MC (M3534, 2.5x magnification). All experiments (except targeting of INPP4B) have been performed in 2 independent series using 2 shRNAs per target. Statistically significant differences to the Scramble controls were calculated by using 1-way ANOVA (**B, D, G, J, M, Q**) and 2-way ANOVA (**C, F, I, L, O**) followed by Dunnett's post test to determine significance and data are represented as mean ± SEM (**p* < 0.05; ***p* < 0.01; ****p* < 0.001; *****p* < 0.0001). For bar- and dot-plot figures, source data are provided in the Source Data file.

---

significantly influenced by the biased targeting of cycling myeloid progenitors by the murine stem cell virus (MSCV) used for transduction but also by a permissive chromatin conformation associated with myeloid differentiation[56,57]. Reconstitution experiments of flow-enriched and virally transduced cells showed that rKMT2A-MLLT3 can initiate AML from the HSC compartment resulting in a more aggressive disease with tumor cells expressing Evi1[6]. Similarly, transplant of naïve iKMT2A-MLLT3 LT-HSC in DOX-treated recipients induced in few recipients a particularly rapid and invasive Evi1⁺AML in a however unpredictable manner and it remained unclear whether the disease originated from LT-HSC or MPP1[9]. The higher fraction of active cycling cells in MPP1 suggested that exogenous factors inducing LT-HSC proliferation may increase the permissiveness for KMT2A-MLLT3 transformation. In contrast to poly-IC or 5-FU, a single dose of TPO significantly and specifically increased the fraction of cycling Evi1^high LT-HSC. This effect was reflected by a strong correlation between expression of the TPO receptor Mpl and Evi1. Although a positive correlation between MPL and EVI1 mRNA expression in AML patients was previously proposed, it has not been experimentally substantiated in an AML mouse model driven by retroviral Evi1 overexpression[58]. Earlier ChIP experiments found binding of overexpressed EVI1 to the MPL gene locus in human solid cancer cells[59]. In addition, germline MECOM mutations associated with BM failure and thrombocytopenia were able to regulate MPL promoter sequences in reporter assays in human cell lines[60]. However, Mpl was not found as a target of a virally overexpressed EVI1-MDS1 fusion in mouse haematopoietic cells[42,61]. We observed that EVI1 KD did not change MPL expression in KMT2A-rearranged human OCI-AML4 cells. In addition, exposure of the cells to TPO resulted in STAT5 activation but not in higher EVI1 expression, which does also not support direct transcriptional regulation in these cells. Our observation therefore suggests that TPO/MPL signaling induces proliferation of EVI1⁺HSC without regulating EVI1 expression.

A single dose of exogenous TPO accelerated induction of iKMT2A-MLLT3 AML originating from HSC. TPO exposure affected the expression signatures of DOX-induced iKMT2A-MLLT3 HSC as early as 2 days, but also in emerging AML cells, with increased expression of genes related to TPO/MPL signaling pathways. Hereby particularly significant correlations to expression profiles from mouse and human AML cells with aberrant HOXA9 expression became evident. Earlier work suggested that TPO/MPL activation results in increased transport of Hoxa9 into the nucleus of murine Ba/F3 pro-B cells and in primary HSPC[55]. Likewise, we observed that TPO stimulation increased the amount of nuclear HOXA9 in EVI1⁺ KMT2A-r human OCI-AML4 cells, which was accompanied by an increase in pSTAT5. Functional cooperation of HOXA9 with STAT5, one of the central transcriptional TPO-MPL signaling effectors, was recently reported in T-cell acute lymphoblastic leukemia[62]. Although no physical interaction between STAT5 and HOXA9 has ever been demonstrated, co-occupancy on chromatin was associated with potent up-regulation of multiple genes

downstream of JAK/STAT signaling. Although still at the early phase, combination of future small molecules that allow to efficiently and specifically interfere with aberrant HOXA9/STAT5 signaling could therefore also be efficient in HSC-derived EVI1⁺ AML[63,64].

Comparing gene expression signatures of iKMT2A-MLLT3 mouse AML with human AML expressing elevated EVI1 levels revealed several common DEG. Notably, many of the higher expressed genes are also highest expressed in normal HSC such as IL12Rβ2 encoding for a subunit of the IL12 and IL35 receptor complex. IL-12 is primarily known as a regulator of cell-mediated immunity rather than a central HSC regulator[65]. However, IL-12 was earlier shown to increase HSC numbers, at least in vitro, suggesting that, similarly to TPO, IL12 may affect cycling of EVI1⁺ HSPC and increase susceptibility for transformation by KMT2A-r[49]. In addition, while in vivo administration of IL12 decreased BM haematopoiesis via IFN-γ induction[66], it facilitated the recovery and engraftment of HSC after radiation. Interestingly, Il12rβ2 was very recently found to be significantly expressed in foetal liver HSC as early as E14 in normal mice, suggesting an important role for hematopoietic development[67]. IL12Rβ2 also appeared in earlier potentially prognostic leukemic stem cell gene expression signatures[68,69]. We observed that IL12Rβ2 KD impaired the colony formation not only in EVI1⁺ OCI-AML4 cells but also in EVI1⁻ MOLM13, THP1 and HL60 cells, indicating a potential role as driver and putative therapeutic target in AML. However, the fact that the cells grow in IL-12-free medium suggests a yet unknown mechanism of action.

We and others have shown that EVI1⁺ HSC-derived AML is generally associated with a more aggressive biology with chemoresistance and poor outcome. Although exogenous factors have been discussed to eventually increase the susceptibility of HSC-derived AML, they were never functionally explored. Our findings suggest a model in which exogenous stimuli like aberrant levels of a cytokines such as TPO specifically increase the numbers of cycling EVI1⁺ HSC (LT-HSC, MPP1) which raises the susceptibility for transformation by the iKMT2A-MLLT3 fusion oncogene. This finding may have some wider implication as EVI1-expressing AML lacking 3q26 lesions are not limited to those carrying KMT2A-rearrangements. It will be therefore interesting to see whether increased cycling of EVI1⁺ HSC is also a prerequisite to develop EVI1⁺ AML driven by lesions like NUP98-r or mutant NPM1 found in some patients. Aberrantly high TPO levels as observed in patients developing rebound thrombocytosis after e.g., induction chemotherapy or in other diseases, or in MDS patients treated with TPO agonists may result in significantly increased cycling of EVI1⁺ HSC[70,71]. However, the fact that EVI1⁺ AML without 3q26 lesions are relatively rare suggests that the risk to develop a potent driver lesion like a KMT2A-MLLT3 fusion in TPO-stimulated cycling EVI1⁺ HSC is very low. Hence, multiple studies did not find any statistically significant evidence for increased risk to develop AML after treatment for MDS-associated thrombocytopenia with the TPO agonist romiplostim[72]. Future work is needed to identify and validate additional exogenous

factors that increase HSC cycling and the susceptibility for malignant transformation by drivers beyond KMT2A-MLLT3. The identification of these cellular mechanisms is essential to develop therapeutic strategies that target the origin-related gene expression programs in AML cells that contribute to stemness, drug resistance and disease relapse.

## Methods

The authors state that this research complies with all relevant ethical regulations. All animal experiments were carried out in adherence to the Swiss law for animal welfare and approved by the Swiss Cantonal Veterinary Office of Basel-Stadt, Switzerland, under the animal license number 2087-31838 (to J.S). In all experiments, the maximal allowed severity degree did not exceed. Mice were generally used between 6–15 weeks of age. For in vivo experiments, we transplanted cells from female donors into female recipients.

### Transgenic mice

iKMT2A-MLLT3 mice[9] were crossed with the Evi1[+/GFP] transgenic strain[16], leading to the generation of the iKMT2A-MLLT3;Evi1-IRES-GFP line ("KME"). The mice were kept under specific pathogen-free conditions at the animal facility of the Department of Biomedicine (University of Basel, Basel, Switzerland) with free access to food and water in accordance to Swiss Federal Regulations. Mice were genotyped using the AccuStartII Mouse genotyping kit (Quanta Biosciences, Beverly, MA, USA) for extraction of genomic DNA and PCR reaction. The AccuStartII GelTrack PC SuperMix was used for PCR amplification. PCR amplicons were visualized on 2% agarose gels containing 2–4% Ethidium Bromide (Cat.15585011, Invitrogen, Thermo Fisher Scientific, Waltham, MA, USA). Primers are provided in Supplementary Data file 11.

### Exogenous stimuli

5-fluorouracil (5-FU, Cat.F-6627, Sigma-Aldrich, Merck, Darmstadt, Germany), polyinosinic:polycytidylic acid (poly(I:C), Cat. P1530, Sigma-Aldrich, Merck), thrombopoietin (TPO, murine TPO, Cat. 315-14, PeproTech, Thermo Fisher Scientific) and Romiplostim (RP, Nplate®, Amgen, Thousand Oaks, California, USA) were injected i.p. in single doses: 150 mg/kg for 5-FU, 10 mg/kg for poly(I:C), 200 µg/kg for TPO and RP. Mice were analyzed 3 and 6 days after injection for the 5-FU treatment. For the poly(I:C)-treated animals, they were analyzed 24 h post-injection, while the TPO- and RP-treated mice were analyzed after 48 h.

### Colony forming assays

Flow cytometry sorted LT-HSC or MPP1 cells were plated at a concentration of $10^4$ cells in 2 ml of MC M3534 for mouse cells, H4534 for human cell lines (Methocult™, STEMCELL Technologies, Vancouver, Canada) with 1µg/ml DOX (Cat.631311, Clontech, Takara Bio, Mountain View, California, USA) in a 35 mm Nunc™ dish with 2x2mm grid (Cat.174926, Thermo Fisher Scientific). Uniform plating was achieved by using 37 °C pre-warmed MC and resuspending cells in a 2 ml syringe (CODAN, Lensahn, Germany). Colonies were incubated for 8–10 days at 37 °C until scored. Scoring was performed on a Zeiss Axiovert 40 CFL microscope (Zeiss, Jena, Germany). Pictures were taken on a Zeiss Vert.A1 inverted microscope with 2.5x and 5x objectives. Cells were washed twice in warm PBS, and counted with trypan blue (Cat.T8154, Sigma-Aldrich, Merck). Cells were then analyzed by flow cytometry. For propagation, cells were plated at a concentration of $10^5$ cells per dish, in fresh MC and DOX.

### BM transplantation

Flow cytometry-sorted progenitor cells from KME mice (CD45.2) were tail-vein injected in lethally irradiated (2x 6 Gy, Gammacell® 40 Exactor, Best Theratronics, Ottawa, Canada; Cesium source) 8-10 weeks-old B67.SJL/Crl (CD45.1) mice with $2 \times 10^6$ supporting normal BM cells of

B6.SJL/Crl donor mice. DOX was provided in water (500 µg/ml) or as impregnated food pellets corresponding to 1.2 mg/day (400ppm DOX diet, Envigo, Inotiv, West Lafayette, Indiana, USA). Cellular chimerism was assessed by flow cytometry in peripheral blood collected using standard procedures every 3 weeks post-transplant.

### Analysis of DOX-induced or transplanted mice

Mice were sacrificed once symptomatic by $CO_2$ asphyxia in accordance with Swiss regulations. Organs were collected, weighted and stored in 4% Paraformaldehyde solution for further histological analysis. Blood, BM and spleen were counted and analyzed by flow cytometry as described below. Additionally, each blood sample collection was accompanied by whole blood cell measurement on a blood cell counting machine (ADVIA 2120i Hematology System, Siemens Healthineers, Erlangen, Germany). Paraffin-embedded tissue sections were stained with hematoxylin and eosin. Morphology was analyzed on smears stained with Wright-Giemsa staining (Hematology, University Hospital, Basel, Switzerland). Sections were imaged on a Nikon TI microscope (Nikon, Tokyo, Japan) and analyzed using Fiji ImageJ software (version 2.3.0/1.53q)[3].

### Analysis of mouse haematopoietic cells

Total BM was harvested by crushing long bones and spine in RPMI1640 (Cat.61870044, Gibco, Thermo Fisher Scientific) containing 10%Foetal Calf Serum (FCS, Cat.2-01F10-I, BioConcept, Allschwill, Switzerland) and 1%Pencillin/Streptomycin (P/S, Cat.15140122, Gibco, Thermo Fisher Scientific) and then filtered through 40 µm cell strainer (Cat.7.542.000, Greiner Bio-One, St-Gallen, Switzerland). Spleens were dissected, weighted, and single cell suspensions obtained by pressing through a 40 µm cell strainer. Red blood cells were lysed with ammonium-chloride potassium (ACK) lysis buffer (150 mM $NH_4Cl$, 10 mM $KHCO_3$, and 0.1 mM EDTA, pH8.0) for 10 min on ice. After a 5 min centrifugation at $250 \times g$ (Heraeus Megafuge 8 Centrifuge, Thermo Fisher Scientific), cells were resuspended in RPMI1640 with 10%FCS and 1%P/S. Lineage marker depletion was achieved according to manufacturer's protocol of Mouse Hematopoietic Direct Lineage Depletion Kit (Cat.130-110-470, Miltenyi Biotech, Bergisch Gladbach, Germany). Cytospins of approximately $2 \times 10^5$ cells were made by centrifugation for 6–7 min at $12 \times g$ using a Cytospin-4 centrifuge (Thermo Fisher Scientific) with cytofunnel disposable sample chambers (Single Cytofunnel with white filter cards, Cat.5042000, Thermo Fisher Scientific) and non-coated cytoslides (Cat.5991051, Thermo Fisher Scientific). Cytospins were stained with Wright-Giemsa solution, imaged on a Nikon TI microscope (Nikon) and analyzed using Fiji ImageJ software (version 2.3.0/1.53q).

### Flow cytometry

Cells in suspension were washed with FACS buffer (0.5% BSA, 1 mM EDTA in PBS, filtered) and incubated with the indicated antibodies for 1 h at 4 °C in 100 µl FACS buffer. After staining, cells were washed and resuspended in 200 µl FACS buffer. Stained cells were analyzed on a LSR Fortessa (BD, New Jersey, USA). Data was analyzed with FlowJ™ v10.8 Software (BD Life Sciences). For stem and progenitor staining, lineage positive cells were depleted as described in the previous paragraph. For fluorescence activated cell sorting, a BD FACS ARIA III (BD) was used, and cells were collected in 1.5 ml Eppendorf tubes with medium. All antibodies and used dilutions are provided in Supplementary Data file 12.

### Cell cycle analysis

HSPC staining was performed for 1 h at 4 °C. Anti-GFP antibody (Rockland Immunochemicals, Limerick, PA, USA) at 1:200 dilution was included in the staining mix for this experiment. Cells were resuspended in 50 µl of cold PBS and quickly vortexed. Cell suspension was incubated in 300 µl of 4x diluted BD Cytofix/Cyotperm solution

(Fixation and Permeabilization solution kit, Cat.554714, BD Biosciences, BD) for 20 min on ice. Cells were then washed twice with 1 ml of room temperature BD Perm/Wash Buffer (Fixation and Permeabilization solution kit), centrifuged at $400 \times g$ for 5 min each time to remove the supernatant, before a final resuspension in 300 μl BD Perm/Wash Buffer with Ki-67 antibody (1:30-1:100 dilution) and overnight incubation at 4 °C in the dark. The following day, cells were washed in 1 ml BD Perm/Wash Buffer and incubated for 10 min on ice in a solution of BD Perm/Wash Buffer with a concentration of 4 μl of 1 mM DAPI stock solution (Cat.D9564, Sigma-Aldrich, Merck). After a final wash in 1 ml BD Perm/Wash Buffer, the cells were resuspended in 200–300 μl of FACS buffer and acquired on a LSR Fortessa (BD). All antibodies and used dilutions are provided in Supplementary Data file 13.

### Quantitative reverse-transcriptase PCR (qRT-PCR)

Total RNA was extracted according to manufacturer's instruction using the Quick-RNA Microprep or Miniprep kits (Cat.R1050 resp. R1054, Zymo Research, Irvine, California, USA) depending on cell number. cDNA synthesis was made using the high-capacity cDNA reverse transcription kit (Cat.4368814, Applied Biosystems, Thermo Fisher Scientific). Quantitative PCR was performed using either Taq-Man probes (Applied Biosystems, Thermo Fisher Scientific) or SYBR Green reagent (Applied Biosystems, Thermo Fisher Scientific), and an ABI Prism® 7500 sequence detection system (Applied Biosystems, Thermo Fisher Scientific). Ct values were normalized to housekeeping genes as described in the legends, and relative expression was quantified using $1/\Delta Ct$. Primers are listed in Supplementary Data file 14.

### Preparation of retroviral supernatants and knockdown in human and mouse AML cells

Viral supernatants were produced by co-transfection of HEK293T-LX with an ecotropic packaging vector mix (pMD2G envelope plasmid, pMLDg/PRE packaging plasmid and pRSV/Rev Reverse transcriptase plasmid) and the pLT3-P2A-mCherry knockdown plasmid[73] carrying the respective mir-shRNA by using Jetprime transfection reagent (Cat.R101000046, Polyplus, Illkirch, France). Supernatants were harvested 24, 48 and 72 h after transfection before snap-freezing in liquid nitrogen and storage at −80 °C. OCI-AML4 cells, as well as MOLM-13, THP1, HL-60 as well as primary mouse AML cells were transiently spin-infected with supernatants in presence of 7.5 μg/ml polybrene (Cat.107689, Sigma-Aldrich, Merck) for 90 min, 1000 × g at 30 °C. Transduced AML cell lines were incubated overnight, and plated in αMEM, RPMI1640 (Cat.12571063/61870044, Gibco, Thermo Fisher Scientific) with 20% FCS and 1% P/S. Cells were incubated in 2 ml medium with 1 μg/ml Puromycin at 37 °C, 5% $CO_2$ for 48 h. Cells were then grown for a few days before induction of the shRNA-mediated knockdown with 1 μg/ml DOX for 72 h. Cells were then sorted according to their mCherry expression and used for experiments. Primary mouse AML cells were incubated overnight, and plated in Transplant Medium (RPMI1640 with 10% FCS, 1%P/S, 10 ng/ml mSCF, 6 ng/ml IL3 and 10 ng/ml IL6) for 48 h before sorting for double GFP & mCherry positive cells.

### Cell lysate preparation and sub-cellular fractionation

OCI-AML4 cells were starved in serum-free medium overnight before stimulation with 100 ng/ml TPO (or similar volume of PBS as control) for 15 min. The cells were pelleted and the pellets were directly (without PBS washing) resuspended in hypotonic buffer (10 mM Tris HCl pH 7.6, 1.5 mM MgCl2, 10 mM KCl, 1X EDTA-free Roche protease inhibitors). After 10 min of incubation, Triton X-100 was added to a final concentration of 0.3%, which was followed by 10 s vortexing. The lysates were then centrifuged for 1 min at $9500 \times g$. Supernatant corresponded to cytoplasmic fraction, while the pellet was further resuspended in nuclear extraction buffer (20 mM Tris HCl pH 7.6,

420 mM NaCl, 20% glycerol, 2 mM MgCl2, 250U Merck Benzonase, 1X EDTA-free Roche protease inhibitors). Nuclear lysates were incubated for 1 h at 4 °C on a rotating wheel, then centrifuged at $13,200 \times g$ for 30 min. The supernatant was collected and used for further experiments as a nuclear fraction.

### Western blot analysis

Nuclear and cytoplasmic fractions were analysed by 4–12% Bis-Tris Gel (Thermo Fisher Scientific) and transferred to the nitrocellulose membrane (Porablot Nitrocellulose, Cat.71280, Macherey-Nagel, Düren, Germany) by overnight wet transfer (BioRad transfer chamber). The membrane was blocked using 5% milk PBS-T solution (0.1% Tween in PBS) for 1 h at RT. Primary antibodies (Anti-HOXA9, Anti-STAT5, Anti-STAT5P and Anti-GAPDH) diluted in 5% milk solution according to manufacturer's instructions, were incubated for 2 h at RT with the nitrocellulose membranes. Washing of the nitrocellulose membranes was done 3 times by PBS-T for 5 min, after which secondary antibodies (anti-mouse and anti-rabbit) were incubated for 1 h. After 3 more washes by PBS-T, membranes were developed using the SuperSignal™ West Femto Maximum Sensitivity Substrate (Cat.34095, Thermo Fisher Scientific). All antibodies are listed in Supplementary Data file 15.

### RNA sequencing: RNA isolation and library preparation

Total RNA was isolated using the Quick-RNA Miniprep kit (Cat. R1054, Zymo Research). Concentration was measured on a NanoDrop2000 (Thermo Fisher Scientific), confirmed by fluorometry using the Quantifluor RNA system and RNA quality was checked on a Bioanalyzer using the RNA 6000 Nano Chip. Library preparation was performed with 15 ng total RNA using the TruSeq stranded mRNA Library Kit. 10 cycles of PCR were performed. Libraries were quality-checked on the Fragment Analyzer using the Standard Sensitivity NGS Fragment Analysis Kit.

### RNA sequencing: data analysis

RNA-seq paired-end reads were mapped to the mouse genome assembly, version mm10, with STAR[74] (version 2.7.9), with default parameters except for using the multi-map settings (outFilterMultimapNmax=10 and outSAMmultNmax=1) and filtering reads without evidence in spliced junction table (outFilterType = "BySJout"). The output bam files of the same sample sequenced on different sequencing lanes were merged, sorted and indexed with SAMtools[75] (version 1.15). The featureCounts[76] software of Subread[77,78] (version 2.0.1) was used to count the number of read (5'ends) overlapping with the exons of each gene assuming an exon union model based on the genes annotation provided by Ensembl database (version 102). Counts were normalized in log Count per Million (CPM) and used for the gene expression plots as well as principal analysis complement (PCA). Differentially gene expression analysis was performed with the EdgeR[79] (version 3.36) Bioconductor package. The design to analyze the data was: -0 + group, where group corresponds to the groups classified according to fusion Mecom expression and the TPO treatment.

For the human datasets, RNAseq datasets were downloaded from the TARGET[46] and BEAT[47] databases with the R package TCGAbiolinks (version 2.28.4)[80,81]. For TARGET database, the results published here are in part based upon data generated by the Therapeutically Applicable Research to Generate Effective Treatments initiative, phs000218 (https://www.cancer.gov/ccg/research/genome-sequencing/target). AML and LAML samples were filtered from "TARGET-AML" (phs000465) and "BEAT-AML1.0-COHORT" (phs001657) projects based on supplemental information about patients available at the Genomic Data Commons website (https://portal.gdc.cancer.gov). For the Leucegene database (https://leucegene.ca/), the human samples from the dataset GSE67040 were downloaded from the gene expression omnibus website (https://www.ncbi.nlm.nih.gov/geo/)[82]. For the ST.JUDE datasets[48], the feature count files of the RNAseq datasets from

353 pediatric patients were obtained from ST.JUDE Cloud[83]. The accession number of used datasets are SJC-DS-1013 (PedAML) and SJC-DS-1021 (PanpAML). Log2 TMP or RPKM (for the Leucegene or ST.JUDE datasets) expression for the MECOM and ERG genes were used to group samples in 4 or 5 clusters (5 clusters only for the ST.JUDE dataset) with the k-means function available in the R package stats with the following parameters: center = 4 (or 5 for St-Jude), nstart = 25. The 2 clusters of human samples with the higher levels of expression for MECOM and ERG were selected, and compared by a differential expression analysis based on their raw gene expressions with the R package EdgeR.

## Single-cell RNA sequencing: experimental design

Only female mice were used to eliminate any sex-related effects. Similarly, to remove any possible effect of the DOX-food on the analysis, all animals were given DOX-infused pellets. Control animals were chosen based on genotype.

Lin-BM cells from KME mice subjected to TPO or PBS injection followed by DOX-food for 2 (4 replicates) or 5 days (3 replicates), were stained for FACS sorting of LSK Flt3- population. Additionally, CITE-seq HTO labeling (BioLegend, San Diego, CA, USA) of LT-HSC was added to the staining mix for 1 h at 4 °C. Each sample was also labeled individually with a different CD45 Hashtag HTO antibody (BioLegend), for in silico identification of the different conditions. After washing, cells were resuspended in FACS buffer and sorted on a SORPAria (BD). The sorted cells were counted and assessed for viability, then processed together for scRNA-Seq according to manufacturer's recommendation (10x Genomics, Pleasanton, CA, USA). For preparation of HTO-surface libraries manufacturer's recommendations (BioLegend) were followed. All antibodies for FACS, ADT ( = CITE-seq) and HTO are listed in Supplementary Data file 16.

## Single-cell RNA sequencing analysis

Single-cell capture, cDNA and library preparation were performed at the Genomics Facility (D-BSSE, ETH Zurich, Basel, Switzerland) with a 10x Genomics Single Cell 3' v3 Reagent Kit. BioLegend TotalSeq-B antibodies were used to produce hashtag oligos (HTO) and antibody-derived tag (ADT) libraries. Sequencing was performed on three flow-cells of GFB Novaseq 6000 instrument resulting in 90/28nt- and 101/28nt-long paired-end reads.

The dataset was prepared and sequenced in two batches (corresponding to two time-points: day 2 and day 5 after induction; see design single cell RNA sequencing). The two time-points were analysed together by the Bioinformatics Core Facility (Department of Biomedicine, University of Basel, Basel, Switzerland). Read quality was assessed with the FastQC tool (v0.11.5). Individual libraries were aligned to the mouse 'mm10' genome and two additional sequences (the KMT2A-MLLT3 fusion gene and the EGFP Enhanced green fluorescent protein gene) using the STARsolo (https://github.com/alexdobin/STAR/blob/master/docs/STARsolo.md) tool (v2.7.9a) with default parameters except for: soloUMIlen=12, soloBarcodeReadLength=0, clipAdapterType=CellRanger4, outFilterType=BySJout, outFilterMultimapNmax=10, outSAMmultNmax=1, soloType=CB_UMI_Simple, outFilterScoreMin=30, soloCBmatchWLtype=1MM_multi_Nbase_pseudocounts, soloUMIfiltering=MultiGeneUMI_CR, soloUMIdedup=1MM_CR, soloCellFilter=None. Read counting was done using Ensembl 102 gene models.

ADT/HTO libraries were also processed and mapped using the STARsolo tool with default parameters except soloCB-matchWLtype=1MM_multi_Nbase_pseudocounts, soloUMIfiltering=MultiGeneUMI_CR, soloUMIdedup=1MM_CR, soloCellFilter=None, clipAdapterType=False, soloType=CB_UMI_Simple, soloBarcodeR-eadLength=0, soloUMIlen=12, clip5pNbases = 10, clip3pNbases = 76 (for 101nt reads) or 65 (for 90nt reads).

Further analysis steps were performed using R (v4.2.0) and mostly following the steps of the workflow presented at https://bioconductor.org/books/3.15/OSCA/. The emptyDrops[84] function from the DropletUtils package (v1.16.0) was used to remove empty droplets (UMI = 0) and droplets containing only ambient RNA (FDR threshold < 0.001). Default parameters were used (except niters=5000). Genes high in ambient (proportion > 0.001) were tagged and excluded from the highly variable genes later on in the analysis. The hashedDrops function from the DropletUtils package was used to demultiplex cell barcodes into their samples. Droplets for which the demultiplexing failed were removed.

From the gene expression libraries, cells were considered of high quality if UMI counts >= 2511, genes detected >= 1250 and mitochondrial proportion >0 and =< 10%, forming a final dataset of 108,503 high-quality cells. These thresholds were defined using the observed distributions across all cells and samples.

The filtered count matrix was normalized using a deconvolution strategy[85]. Cells were pre-clustered using the quickCluster function from the scran[86] package (v1.24.0). Sample-specific effects were removed using the fastMNN[87] function (d = 50, k = 50) from the batchelor package (v1.12.1). Clustering was performed using FindNeighbors (dims=1:20, k = 20) and FindClusters (resolution=0.6) functions from the Seurat[88] package (v4.1.1). The cluster "0" was refined using FindNeighbors (dims=1:20, k = 20) and FindClusters (resolution=0.3) functions resulting in three sub-cluster ("0.0", "0.1" and "0.2"). Doublets were identified and removed using scDblFinder from the scDblFinder[89] package (v1.10.0). Doublets were searched independently for each sample as they represent different captures. Initial analysis defined 22 clusters; however, 4 clusters were removed as they could not be investigated for differential expression for lack of sufficient cell numbers or bad quality cells.

A posteriori gating of cells to the LT-HSC, ST-HSC, MPP0, MPP2 and MPP3/4 subpopulations was performed based on the surface protein signal from the CITE-Seq antibodies following the gating strategy by Fast et al.[34].

The clusters were initially annotated by using singleR with the Immgen microarray coarse data and the gating results. The final annotation was done based on the expression of proposed marker genes[30,90,91].

Cell cycle phases were assigned to cells using the cyclone[4] function from the scran package (v1.24.0)[92]. The dataset was also subjected to cell-type annotations using the package SingleR[93] (v1.10.0). Only high-quality assignments (pruned scores) were used. Single-cell RNA-seq transcriptional profiles of mouse LSK progenitors were used as reference data set[39].

A T-SNE dimensionality reduction was used for visualizing single cells in two dimensions. The T-SNE coordinates were calculated based on the sample-corrected low-dimensional coordinates for each cell using the runTSNE function from the Scater[94] package (v1.24.0) with default parameters.

Pseudo-bulk differential expression analysis[5] was performed using the edgeR package (v3.34.1). Differentially expressed genes were detected by likelihood ratio tests[95] (FDR < 0.05). Pseudo-bulk aggregation was performed only for cells from the same cluster and sample, if the number of cells in the group was at least 20[96]. Gene set enrichment analysis was performed using the camera function from the limma (v3.52.2) package with gene set collections from the MSigDB[97,98] database (v7.5.1).

## BM slice preparation, immunostaining and optical clearing

Methods for 3D imaging of BM were adapted from published protocols[26]. Mouse femurs were isolated, cleaned and immersed in 2% paraformaldehyde (PFA) in PBS (Pierce™ 16% Formaldehyde, Methanol-free, Cat.28908, Thermo Fisher Scientific) for 6 h at 4 °C, followed by a dehydration step in 30% sucrose for 72 h at 4 °C. Femurs were then embedded in cryopreserving medium (optimal cutting temperature (OCT), Leica Biosystems, Nussloch, Germany) avoiding the appearance

of air bubbles and immediately snap frozen in liquid nitrogen. Samples were kept at −80 °C for long-term storage. Bone specimens were iteratively sectioned using a CM3050 S cryostat (Leica Biosystems) until the BM cavity was fully exposed along the longitudinal axis. The OCT block containing the bone was then reversed and the procedure was repeated on the opposite face until a thick bone slice with bilaterally and evenly exposed BM content was obtained. Once BM slices were generated, the remaining OCT medium was removed by incubation and washing of the bone slices in PBS 3 times for 15-20 min. The samples were additionally fixed in 2% PFA/PBS for 2–4 h and washed 3 times for 1 h (washing solution: 0.2% Triton X-100 in PBS). For immunostaining slices were incubated in blocking solution (0.2% Triton X-100, 1% bovine serum albumin (BSA), 10% donkey serum in PBS) overnight at 4 °C. Blocking solution was removed and primary antibodies in blocking buffer solution were added for incubation at 4 °C with agitation for 48 h and washed thoroughly three times with washing solution. Secondary antibody staining was performed at 4 °C with agitation for another 48 h and were washed thoroughly 3 times with washing solution. Femur slices were transferred into Rapi-Clear 1.52 (Cat.RC152001, Sunjin Lab, Hsinchu City, Taiwan) for clearing overnight at 4 °C, which typically increased imaging depth to 150 μm from the tissue surface without significant loss of signal intensity.

## Confocal imaging and microscopy data visualization

For observation under the confocal microscope, BM slices were mounted on glass slides using vacuum grease. Confocal microscopy was performed with 10x (HCX PL FLUOTAR), 20x (HC PL APO CS2) and 63x (HCX PL APO CS2) on an SP5 Leica confocal microscope. Low-magnification tissue-wide images were generated and ROIs were defined for subsequent detailed acquisition with 20x and 63x. Image analysis was exclusively performed in high-resolution images, which met quality standards. In this microscope, 405 nm diode laser and a white light laser (470–670 nm) tuned to specific wavelengths was employed. Fluorescence was detected using ultrasensitive Leica Hybrid detectors (HyD). Attenuation of fluorescent signal with tissue depth was minimized using the tool for laser intensity correction along the z-axis available on the Leica Application Suite X. 3D whole-mount images were analyzed using the commercially available software Imaris v10.2 (Bitplane AG, Oxford Instruments, Abigdon-on-Thames, UK). A median filter (3 × 3 × 3 pixels) was applied to channels containing HSC and progenitor labeling in order to reduce pixelate noise and simplify manual cell annotation.

## Volumetric quantitative image analysis

Double-positive (Evi1$^+$-Ckit$^+$) HSC cells were manually annotated using the spots module of Imaris software (v10.2 Bitplane AG, Oxford Instruments) to ensure a minimal error rate. The segmentation of the DAPI-positive tissue regions and vascular structures was performed using a deep learning-based segmentation tool integrated within a graphical user interface, capable of directly processing microscopy data in Imaris IMS format. The tool employs a modified U-Net convolutional neural network architecture, configured and trained using the nnU-Net pipeline[99]. This approach utilizes the DAPI channel, marking nuclei, along with a secondary vessel marker channel (endoglin or endomucin) to generate precise volumetric segmentation masks. Training data consisted of manually annotated microscopy scans from metaphyseal and diaphyseal regions of femur BM. To enhance generalization, synthetic augmentation incorporating realistic noise distributions was applied to the dataset. Images were processed in patches, with individual patch segmentations subsequently merged to yield continuous segmentation masks, enabling accurate quantification of total tissue volume and detailed vascular structures.

## Data and statistical analysis

Bar graphs represent the mean value of biological replicates, with error bars as the standard error of the mean (mean±SEM). Statistical significance was tested with unpaired t-test, unless otherwise stated. For flow cytometry, statistical significance tests were performed in Prism9 (GraphPad Software, Dotmatics, Boston, MA USA) in $\log_{10}$, for qPCR in $\log_2$ and for colony counts in linear space. Flow cytometry analysis was performed using FlowJo™ v10.8 Software (BD Life Sciences, Franklin Lakes, NJ, USA).

## Illustrations

The schematics (Figs. 2D, 3A, Supplementary Fig. 1A, Supplementary Fig. 4A) included modified illustrations from the Noun Project under Creative Common license generated by Pryanka (Blood), Phillip Glen (Bone), Ade Nur Hidayat (Cells), Turkkub (Laser), Icon Lauk (Syringe). The mice are home-drawn inspired by H.Alberto Gongora.

## Reporting summary

Further information on research design is available in the Nature Portfolio Reporting Summary linked to this article.

## Data availability

The mouse RNAseq and scRNAseq data generated in this study have been deposited and are publicly available in the GEO database under the accession numbers GSE261134 and GSE261266 respectively. Source data are provided with this paper.

## Code availability

R code used for the RNA-seq data analysis is available on Zenodo at https://doi.org/10.5281/zenodo.16845287. R code used for the scRNA-seq data analysis and the figures creation is available on Zenodo at https://doi.org/10.5281/zenodo.17525232 and https://doi.org/10.5281/zenodo.17535032 respectively.

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

## Acknowledgements

This work was supported by grants from the Swiss National Science Foundation (31003A_173224/1) and the Stiftung für Krebskranke Kinder Basel (#2022-F006) to J.S. and a Consolidator Grant from the European Research Council (ERC-Co2019 865803) to C.N.-A. We thank Simón Méndez-Ferrer (Cambridge), Antoine HFM Peters (Basel) and Michael D. Milsom (Heidelberg) for their inputs. We thank the animal experimentation- and the flow cytometry facility from the Department of Biomedicine at the University of Basel for their great service. We thank the Bioinformatics Core Facility and, in particular, Florian Geier and Julien Roux for their valuable inputs. The authors also thank the LEUCEGENE group supported by Genome Canada and Genome Québec, who allowed us to use data from human AML specimens provided by the "banque de cellules leucémique du Québec" (https://bclq.org/) (Montreal, Canada).

## Author contributions

Conceptualization: H.E.C.S. and J.S.; Methodology: H.E.C.S., M.A., C.N.A., J.S.; Experimentation: H.E.C.S., S.J., A.L.P., J.Se., J.S., A.E.T., Z.J., F.V., R.S., P.B.; Software & Data Curation: J.Se., A.E.T.; Writing/Original Draft: H.E.C.S. and J.S.; Funding Acquisition: J.S.; Resources: F.O.B., M.A., A.T., W.T., C.N.A, M.K.; Visualization: H.E.C.S.; Supervision: J.S.

## Competing interests

The authors declare no competing interests.
