## [Transparent Peer Review file · Nature Communications]

Thrombopoietin increases the susceptibility for EVI1+ KMT2A-MLLT3-driven AML expressing stem cell genes linked to poor outcome

Corresponding Author: Professor Juerg Schwaller

Version 0:

Reviewer comments:

Reviewer #1

(Remarks to the Author)

This manuscript is based on the analysis of an Evi1-IRES-GFP reporter crossed onto the inducible iKMT2A-MLLT3 AML mouse model, and the effect of TPO stimulation on HSC subpopulations and transformation by the KMT2A-MLLT3 oncogene. TPO treatment caused an increase in cycling and expansion of Evi1 expressing LT-HSC that was associated with increased transformation by KMT2A-MLLT3 leading to more aggressive AML in transplanted mice. TPO appeared to specifically act on Evi1-expressing HSC rather than causing increased Evi1 expression in the HSC. Single cell gene expression analysis was used to show that in the early stages of KMT2A-expression in Evi1^{high} HSC, following induction of the transgene, TPO stimulation resulted in induction of the expression of genes downstream of the Mpl receptor and oncogenic programs downstream of the KMT2A-MLLT3 fusion. Further bulk gene expression analysis and comparison to gene expression profiles in EVI1-expressing human AML revealed a set of genes whose expression correlated with worse overall disease survival. Knockdown of two of these genes, EVI1 (MECOM) itself and IL12RB, resulted in selection against knockdown cells and colony formation in an EVI1^{high} KMT2A-rearranged human AML cell line.

This study provides a fascinating example of how a brief exposure of HSC to specific growth factors can lead to expansion of an HSC sub-population that is susceptible to transformation into a particularly aggressive leukaemia.

Comments:

1. Is the consequence of TPO treatment on LT-HSC and MPP1 cycling and expansion the same in Evi1-IRES-GFP mice to that in uninduced (- Dox) iKMT2A-MLLT3 mice (Figure 1)? This is important to consider since low levels of 'leaky' KMT2A-MLLT3 expression may not be sufficient for transformation but may be sufficient to alter HSC function.

2. The functional validation of MECOM and IL12rB2 knockdown was examined in only 2 human AML cell lines. The lack of MECOM-expressing MLL-rearranged AML cell lines may limit this analysis. However, the specificity of the data would be reinforced by addition of more MECOM-negative cell lines to the analysis. Alternatively, could knockdown be performed in TPO-treated Evi1^{high} iKMT2A-MLLT3 mouse cells, to examine colony formation? Furthermore, at least two different shRNA should be used for each target gene, or an alternative targeting strategy.

Minor points

The sequence of Dox and TPO treatment of the iKMT2A-MLLT3 mice is not clear in some experiments. For example, how long were the mice treated with Dox before exposure to TPO in Figure 2?

Figure 1: the order of TPO treatment is not the same in Fig1-I as that in Fig1-J and K. This is also the case for Fig2-A and Fig2-B.

(Remarks on code availability)

Reviewer #2

(Remarks to the Author)

The authors aim to identify the cell of origin of EVI1+ KMT2A-rearranged AML and characterize the underlying mechanism of disease evolution. The work offers limited advance to the field as a previous study by the same group (Stavropoulou et al, Cancer Cell 2016) reported LT-HSCs as the population that initiates MLL-AF9-driven leukemia. The current study is focused on a subgroup of this type, EVI1+ KMT2A-rearranged AML.

Strengths

The authors made an effort to employ a wide array of tools to address this question, by using transgenic mouse lines, performing scRNAseq, analyzing patient databases and performing in vitro and in vivo experiments to validate and substantiate the findings.

Concerns

The major weakness of the study is that it falls short in its aim to find the cellular origin of EVI1+ KMT2A-rearranged AML and characterize this population in depth. There are several tools available to achieve this, such as single-cell lineage tracing mouse models (CARLIN/DARLIN). In addition, all data are obtained using a single cell line. To show the relevance of the MECOM signaling pathway to iKMT2A-MLLT3 related AML, MECOM should be silenced in the mouse model of iKMT2A-MLLT3 fusion. Trajectory inference analysis using the scRNAseq data, which the authors did perform but not taking advantage of its full potential, could be applied to trace back the originating clone.

Some experiments are poorly designed, and several figures lack data, appropriate controls, normalization/quantification, and/or statistical significance/power. Nomenclature of genes and proteins should be consistent throughout the manuscript. The discussion reads very descriptive and is a missed opportunity to put the results in perspective and inform the readership on the need to identify better, more accurate biomarkers and druggable targets in AML. It also fails to describe how these findings will move the field forward.

- Fig 1: Evi1+ cells in MPP4 as well as CMP, GMP and MEP are missing.

Fig1G does not support the claim that TPO increases Evi1+ HSPCs (no significance and 2 PBS replicates with great variability). A more accurate and robust way to measure this population would be flow cytometry.

Fig1O: The decrease in Mpl expression upon TPO injection is an expected result? It seems counterintuitive and some explanation to this would be helpful.

- Fig2A&B: a reference to this type of colony classification according to their invasiveness phenotype is missing. Also, the Evihigh MPP-derived blasts resulting from TPO injection look more aggressive and numerous than the LT-HSC derived. Could it be that panels have been mislabeled?

Fig2C&D are misnamed in the manuscript (line 157). Fig1D is missing the baseline survival of iKMT2A-MLLT3 AML. The data on 2E and 2F do not show splenomegaly.

Suppl. Fig2C-D could use larger magnification and marking the leukemic blasts with arrows. Increase in Gr1*CD11b+ cells is a hallmark of MLL-AF9 AML, what is the relevance of an increase of Fc RII/III and why is this important in the characterization of the leukemic phenotype?

Data shown in Fig 2H do not appear to be strong enough (25% over half of the recipients, when the control -PBS- shows 1/3). Also, it's not clear what the X axis in the plot is depicting.

Fig2I& J are missing the correlation coefficient (line 172 states a p-value).

- Fig3 What is the rationale to exclude Flt3+ LSK iKMT2A-MLLT3 cells? This experiment is missing a fundamental t=0 control.

Annotation of the populations' cell identity is missing and ADT data are not shown. What is the rationale for performing CITEseq in this experiment?

Administration of DEG 5 days following Dox seems to have similar effect as its administration 2 days following Dox. Increase of Cd74 and Socs2 is mentioned, as well as of Pbx3, Prmt6, Fos and Junb, but these data are not shown in the figure. The figure also lacks p-values for the GSEA signatures on panels H and I.

- Fig4.- It is unclear what panel B represents. What does norm cpm represent and how was it calculated? Erg expression in relation to Evi1 expression is shown but there is no rationale for introducing Erg in this analysis. In addition, there does not seem to be any clear differences among any groups in this panel.

Fig 4G lacks normalized enrichment scores and FDR/ p-values.

- Fig5.- The authors focus on TARGET, the smallest database of the 4 shown in panel 5C, to validate the expression data obtained with the mouse model.

Panel D: Pearson coefficients show a modest correlation between MECOM and IL12Rβ2/INPP4B. The data for the other databases shown in Suppl. Fig.4, which are larger in size, show a weaker correlation.

- Fig6.- For functional validation, silencing of MECOM and its targets is performed in the OCI-AML4 cell line because it expresses high levels of MECOM MPL, IL12Rb2 etc. They show that silencing decreases colony forming units and cell

growth in vitro. However, these events are happening outside the context of the iKMT2A-MLLT3 fusion and in in vitro and therefore they do not prove relevance of the pathway to the disease induced by the fusion. To answer this question shRNA knockdown of MECOM should be performed in the mouse model that expresses the iKMT2A-MLLT3 fusion. Effects on AML burden, and survival should be assessed in vivo.

Supp.Fig.5 shows a western blot that is missing normalization and quantification, as well as controls for the nuclear and cytoplasmic compartments.

(Remarks on code availability)

Reviewer #3

(Remarks to the Author)

In this manuscript, Chatel-Soulet et al seek to clarify the cell of origin of EVI1 expressing KMT2Ar AML using a novel murine model with both inducible expression of the KMT2A-MLLT3 oncofusion and an EVI1 reporter allele. Using this model system they report that administration of exogenous TPO results in a greater fraction of cycling LT-HSC and MPPs and a greater propensity for malignant transformation of these stem/progenitor cells by KMT2A-MLLT3. Gene expression analysis of the resultant murine AML as well as human AML RNAseq data sets identified differential expression of several stem cell associated genes. Knock down of one identified gene, IL12Rb2 (in addition to knock down of MECOM) led to impaired growth of murine and human EVI1+ KMT2Ar AML cells. Overall, the study is well done. The animal modelling is elegant with nice orthogonal validation of some of their findings using human data and a relevant human AML cell line. However a number of issues must be addressed before the manuscript is suitable for publication.

1. Do the authors think the enhanced transforming capacity of HSCs/MPPs after TPO exposure is specific to KMT2A-rearrangement? It seems there are many human AMLs with high EVI1 expression lacking a KMT2Ar, would they postulate other leukemic drivers may likewise have increased ability to transform activated cells after TPO exposure?

2. The mouse experiments and results are poorly described throughout the text making it hard for the reader to follow at times.

a. It is not always clear what comparisons are being made or if these are significant or not. For example, they state in lines 108-110 that TPO led to increased LT-HSC, WBC and trend towards increase BM cellularity, but in suppl fig 1 the WBC and BM cellularity look to be no different than PBS, so are they saying increased compared to the other treatments?

In lines 123-124 it is stated that MPP1 (compared to LT-HSCs) have a 'higher S/G2/M fraction'. While the fraction of MPP1 in S/G2/M is listed, it is not for LT-HSC and based on the figure it also appears these fractions are not significantly different, which is counter to the way it is written in the text. The text states in lines 124-127 that the cycling fraction of LT-HSC and MPP1s increase with TPO, but based on fig 11 it appears this increase is not significant. Such overstatements must be avoided and if the findings are not significantly different should be stated as such.

b. Why in figure J and 1K are the fraction of S/G2/M of both untreated LT-HSC and MPP1s so much higher than that shown in 1I? Were these done under different conditions?

c. In figure 2I and J, counter to the text, it appears there is a correlation between EVI1 expression and disease latency in the mice not stimulated with TPO. What statistical test was applied? The number in the formula for the -TPO is odd ($y = -0.7.442 * X_{86.38}$), what is -0.7.442?

d. The timing of doxycycline induction of KMT2A-MLLT3 relative to the injections of TPO, plpC and 5-FU is not clear in the first three result sections. The results section starting at line 146, is particularly difficult to discern the experiments that are being performed. A description at the beginning of these sections of the experiments being conducted and an experimental schema (like is shown in Fig 3A) is needed.

3. In figure 2G, it is a bit surprising that more of the MPP1 mice with or without TPO are EVI1+ compared to the LT-HSC even with TPO. Can the authors propose some reason for this finding?

4. Why was the differential gene expression analysis of patient data done comparing EVI1 high to EVI1 intermediate whereas the mouse data was EVI1+ vs EVI1 neg? Do the identified gene expression differences replicate if in human data EVI1+ is compared to EVI1 neg?

Minor issues:

There are a few typos:

a) In line 310-311 it says "While induction of INPP4B-targeting shRNAs resulted in cell death (data not shown), it decreased growth and colony formation in MOLM13 cells." Presumably it resulted in cell death in OCI AML4 cells? Please correct this sentence.

b) In line 314 it says "...and expressed higher expression levels...". Remove either expressed or expression from this sentence.

c) In line 349 it states "...activation of STAT5 activation.." remove one activation from this sentence.

d) Please define all abbreviations at first use including KME in line 116 and MC in line 150.

There are also issues with the figures that should be addressed:

a) Figure 1 B – in the text (line 113) it indicates that MPP2 and MPP3 are shown in Figure 1B, but Figure 1B shows MPP1. Please fix the text or the figure.

b) Some of the text is far too small to read. In figure 1 C and D the EVI1 high/low/neg are almost indiscernible. Likewise for EVI1+ and EVI1- in figure 4A, B and E. Same for the labels for the bar graph color schema in Suppl Fig 2H-J and all the font for figure Suppl Fig 4E and F.

c) There is no label for the color scheme in the bar plots for Suppl Fig 1F and G.

d) It appears in the text in lines 246 and 252, figures 5A and 5B, respectively are erroneously called out as 6A and 6B.

(Remarks on code availability)

Reviewer #4

(Remarks to the Author)

I co-reviewed this manuscript with one of the reviewers who provided the listed reports. This is part of the Nature Communications initiative to facilitate training in peer review and to provide appropriate recognition for Early Career Researchers who co-review manuscripts

(Remarks on code availability)

Reviewer #5

(Remarks to the Author)

(Remarks on code availability)

Version 1:

Reviewer comments:

Reviewer #1

(Remarks to the Author)

The authors have addressed all my comments satisfactorily .

(Remarks on code availability)

Reviewer #2

(Remarks to the Author)

The effort the authors have made to correct, amend, and generally enhance the readability and consistency of the manuscript is appreciated.

The primary concerns related to the in vivo experiments necessitated an extended revision period. It is understood that the Carlin experiment could not be conducted, and that silencing MECOM in iKMT2A-MLLT3 transgenic mice has proven technically challenging for the research team. The inclusion of clonogenic assays to reinforce the data derived from primary cells is viewed positively.

Nevertheless, the scRNA-seq analysis can still be improved. While scVelo estimates RNA velocity by measuring the rate and direction of gene expression changes, it does present certain limitations and is somewhat outdated. Therefore, it is recommended that the authors consider applying the Partition-based Graph Abstraction (PAGA) method (Wolf et al., 2019). PAGA offers a simplified representation of cell-type relationships and can effectively identify key transitions between cellular states. This approach has been successfully implemented in studies of the hematopoietic system

Regarding the minor comments, it is suggested that the “custom TPO” GSEA signature be removed from Figure 4H, as its associated p-value does not indicate statistical significance.

(Remarks on code availability)

Reviewer #3

(Remarks to the Author)

This is a nice study with important results and after revisions is much more clearly presented.

(Remarks on code availability)

Reviewer #4

(Remarks to the Author)

(Remarks on code availability)

Reviewer #5

(Remarks to the Author)

(Remarks on code availability)

Version 2:

Reviewer comments:

Reviewer #2

(Remarks to the Author)

The authors have addressed the first comment and have performed the recommended trajectory inference analysis that was the focus of the second comment. The interpretation of this analysis is incomplete, and the comments focus mainly on technical aspects of the analysis and generalized observations. Since TPO induces proliferation of EV1+ HSCs, can this been shown with this analysis since it can be used to track the expression of proliferation genes during the differentiation process.

(Remarks on code availability)

Reviewer #5

(Remarks to the Author)

(Remarks on code availability)

Point to point answers to reviewer's comments

Reviewer #1 - AML fusions (Remarks to the Author):

This manuscript is based on the analysis of an Evi1-IRES-GFP reporter crossed onto the inducible iKMT2A-MLLT3 AML mouse model, and the effect of TPO stimulation on HSC subpopulations and transformation by the KMT2A-MLLT3 oncogene. TPO treatment caused an increase in cycling and expansion of Evi1 expressing LT-HSC that was associated with increased transformation by KMT2A-MLLT3 leading to more aggressive AML in transplanted mice. TPO appeared to specifically act on Evi1-expressing HSC rather than causing increased Evi1 expression in the HSC. Single cell gene expression analysis was used to show that in the early stages of KMT2A-expression in Evi1^{high} HSC, following induction of the transgene, TPO stimulation resulted in induction of the expression of genes downstream of the Mpl receptor and oncogenic programs downstream of the KMT2A-MLLT3 fusion. Further bulk gene expression analysis and comparison to gene expression profiles in EVI1-expressing human AML revealed a set of genes whose expression correlated with worse overall disease survival. Knockdown of two of these genes, EVI1 (MECOM) itself and IL12RB, resulted in selection against knockdown cells and colony formation in an EVI1^{high} KMT2A-rearranged human AML cell line.

This study provides a fascinating example of how a brief exposure of HSC to specific growth factors can lead to expansion of an HSC sub-population that is susceptible to transformation into a particularly aggressive leukaemia.

Comments:

1. Is the consequence of TPO treatment on LT-HSC and MPP1 cycling and expansion the same in Evi1-IRES-GFP mice to that in uninduced (- Dox) iKMT2A-MLLT3 mice (Figure 1)? This is important to consider since low levels of 'leaky' KMT2A-MLLT3 expression may not be sufficient for transformation but may be sufficient to alter HSC function.

We thank the reviewer's overall positive input and opinion about our work. To address the specific points that were raised, we performed additional experiments. We compared the effects of TPO treatment on the cycling and expansion of LT-HSC and

MPP1 in Evi1-GFP and uninduced iKMT2A-MLLT3 mice. We injected mice with a single dose of TPO (i.p., 200mg/kg BW) and sacrificed them 48h later for flow cytometry analysis of the BM cellular fractions. As performed in our original experiments, we used female iKMT2A-MLLT3 mice of similar age (10–11 weeks old) and used PBS-injected mice of the same genotype as controls. As shown in a novel **Suppl. Fig. 1J**, we did not observe any significant differences between the two genotypes.

2. The functional validation of MECOM and IL12rB2 knockdown was examined in only 2 human AML cells lines. The lack of MECOM-expressing MLL-rearranged AML cell lines may limit this analysis. However, the specificity of the data would be reinforced by addition of more MECOM-negative cell lines to the analysis.

We thank the reviewer's input about our functional validation, and we appreciate the comment about the lack of MECOM⁺ KMT2A-rearranged AML cell lines. To address the reviewer's point, we performed additional experiments: We selected 2 additional AML cell lines i) THP-1 cells expressing high levels of *IL12Rβ2* and very low levels of *MECOM*, and ii) HL-60 cells expressing lower levels of *IL12Rβ2* and very low levels of *MECOM*. We focused on MECOM and IL12Rβ2, as INPP4B has been previously characterized in AML. We transduced the cells with shRNA and assessed their growth and colony formation capacity. As expected, expression of MECOM shRNA did not significantly alter growth in liquid cultures or colony formation in methylcellulose. However, rather unexpected, we observed that knockdown of IL12Rβ2 impaired colony formation by MOLM-13, HL-60 as well as THP-1. This data is shown in a revised **Fig.6 F-N**, and described in the MS (**lines353-355**).

Alternatively, could knockdown be performed in TPO-treated Evi1^{high} iKMT2A-MLLT3 mouse cells, to examine colony formation?

We thank the reviewer for this interesting point that we addressed in additional experiments. We also virally knocked down *Il12rβ2* and *Mecom* in Evi1^{high} iKMT2A-MLLT3 AML cells emerging from BM of TPO-treated donors and assessed the impact on colony formation. As shown in new **Fig.6O-R** panels, we observed a significant

reduction of colony formation upon *Mecom* and *Il2rβ2* knockdown mostly affecting type I and type II colonies.

Furthermore, at least two different shRNA should be used for each target gene, or an alternative targeting strategy.

We appreciate the reviewer's comment. We apologize for the oversight. These experiments were performed with at minimum two different shRNA, but as we did not observe any significant differences between most of the working shRNA, we plotted them together. To clarify we now provide the data illustrating the effects of the individual shRNAs in an extended **Suppl. Fig. 6**.

Minor points

The sequence of Dox and TPO treatment of the iKMT2A-MLLT3 mice is not clear in some experiments. For example, how long were the mice treated with Dox before exposure to TPO in **Figure 2**?

To address the reviewer's comment, we added additional schematics that explain the *in vivo* experiments to improve the understanding of our experiments. Specifically, we added a new schematic in **Fig.2D** showing that the mice were treated with TPO 48h before sacrifice, harvest and transplantation of the different cell populations into lethally irradiated mice that were kept on DOX water for 2 weeks, and on DOX food from then on.

Figure 1: the order of TPO treatment is not the same in Fig1-I as that in Fig1-J and K. This is also the case for Fig2-A and Fig2-B.

We thank the reviewer for this comment. To clarify this issue, we modified the graphs that appear now as **Fig. 1H-K** in the revised MS. We also improved **Fig. 2B**.

Reviewer #2 - AML, HSCs (Remarks to the Author):

The authors aim to identify the cell of origin of EVI1+ KMT2A-rearranged AML and characterize the underlying mechanism of disease evolution. The work offers limited advance to the field as a previous study by the same group (Stavropoulou et al, Cancer Cell 2016) reported LT-HSCs as the population that initiates MLL-AF9-driven leukemia. The current study is focused on a subgroup of this type, EVI1+ KMT2A-rearranged AML.

Strengths

The authors made an effort to employ a wide array of tools to address this question, by using transgenic mouse lines, performing scRNAseq, analyzing patient databases and performing in vitro and in vivo experiments to validate and substantiate the findings.

Concerns

The major weakness of the study is that it falls short in its aim to find the cellular origin of EVI1+ KMT2A-rearranged AML and characterize this population in depth. There are several tools available to achieve this, such as single-cell lineage tracing mouse models (CARLIN/DARLIN).

We thank the reviewer for the comment. This was an avenue that we actually thought investigating. However the CARLIN/DARLIN models established by the Carmargo lab depend on a timely-defined window of DOX activity (<https://doi.org/10.1016/j.cell.2020.04.048>; <https://doi.org/10.1016/j.cell.2023.09.019>). The CARLIN model is the most efficient for genetic editing after 7 days of DOX. The DARLIN model, an improved CARLIN version increases the fraction of edited cells with three independent target arrays, but also depends on the limited DOX time window. However, our transgenic iKMT2A-MLLT3 model is fully dependent on constant DOX for transgene expression to induce and maintain the AML phenotype, therefore we cannot use these models.

In addition, all data are obtained using a single cell line. To show the relevance of the MECOM signaling pathway to iKMT2A-MLLT3 related AML, MECOM should be silenced in the mouse model of iKMT2A-MLLT3 fusion.

To address the reviewers points we performed a large series of additional experiments. In a first series of experiments, we used frozen samples from the transplanted mice (Fig. 2I) to obtain Evi1+ iKMT2A-MLLT3+ AML cells and transduced them with the lentivirus (*pLT3*) expressing the respective shRNA together with mCherry. Hereby we observed that the transduction efficacy and viability of the cells was very poor. We therefore decided to generate fresh Evi1-GFP+ iKMT2A-MLLT3+ AML by transplantation experiments in mice. However, due to the ongoing construction of the new University Hospital in Basel (<https://www.unispital-basel.ch/newscenter/campus-gesundheit/20-02-2025>) next to our building, mouse colonies were initially poorly breeding and it took over 3 months to obtain the mice necessary. Nevertheless, we were able to obtain 2 mice with iKMT2A-MLLT3 AML and sufficient Evi1-GFP+ AML cells. To increase the transduction efficacy of the primary cells, we prepared ultraconcentrated (*pLT3*) lentivirus. However again, in two independent experiments, we obtained only very few iKMT2A-MLLT3+ that were GFP+ and also expressed the respective shRNA reflected by the mCherry signal (**Reviewer Fig.1A**). Although the number of cells was not sufficient to transplant a significant number of mice, it nevertheless allowed us to show the KD effects (*shRNA-Mecom*, *shRNA-II12rb2*, *shScramble*) in clonogenic assays (as shown now in the revised **Figure 6O-R**). To increase the number of transduced cells we then explored the use of an old MSCV-based retroviral vector (*pLMP*). We first generated a *pLMP-mCherry* vector and

subcloned the *shRNAs* into it. We then transduced again freshly propagated Evi1-GFP⁺-iKMT2A-MLLT3⁺ AML cells. Hereby we observed that although we significantly increased the transduction efficacy, we again obtained a very low numbers of GFP⁺/mCherry-positive cells, most likely due to the transduction bias of dividing stem- and progenitor cells of the MSCV-based retrovirus (**Reviewer Fig.1B**). Based on these observations we conclude that: to be able to transplant sufficient numbers of primary Evi1-GFP⁺-iKMT2A-MLLT3⁺ AML cells we would need again a significant number of mice to first generate these fresh cells, then transduce them with large amount of ultraconcentrated virus followed by transplantation into a significant number of recipient mice. As we already used here many mice, we cannot obtain sufficient mice on our current license to repeat these experiments within a decent time frame. We agree that these experiments would have been nice but we think that they do not change the overall message of the paper, as along on this path, we performed 4 independent colony-forming assays showing that *Mecom* and *Il12rb2* are important in primary cells.

Trajectory inference analysis using the scRNAseq data, which the authors did perform but not taking advantage of its full potential, could be applied to trace back the originating clone.

We performed a velocity analysis using scVelo within the velocytor package (v1.12) to study cellular differentiation under various conditions (**Reviewer Fig.2A**). Surprisingly, in all conditions, we observed differentiation progressing from more mature to more primitive cell populations, which contradicts the expected direction based on previous studies of the hematopoietic system. A 2022 review (<https://pubmed.ncbi.nlm.nih.gov/34435732/>) highlighted limitations in RNA-velocity modelling for hematopoietic cells, as they exhibit complex gene expression kinetics that can lead to misleading results. These factors may explain the reversed direction in our velocity calculations. Given these concerns, we chose not to include these results in our manuscript.

(A) t-SNE projection of trajectory result done by the velocity analysis.

Some experiments are poorly designed, and several figures lack data, appropriate controls, normalization/quantification, and/or statistical significance/power.

Although the reviewer does not clearly indicate which experiments he is referring to, we tried to overall improve the experiments following his suggestions below:

Nomenclature of genes and proteins should be consistent throughout the manuscript.

We thank the reviewer for comment. Nomenclature has been checked and corrected as required in the revised MS

The discussion reads very descriptive and is a missed opportunity to put the results in perspective and inform the readership on the need to identify better, more accurate biomarkers and druggable targets in AML. It also fails to describe how these findings will move the field forward.

We intentionally tried not to overinterpret our findings. However, to fulfill the reviewers concerns we added a more speculative paragraph to the discussion (**revised MS, lines 460-481**).

- Fig 1: Evi1⁺ cells in MPP4 as well as CMP, GMP and MEP are missing.

We thank the reviewer for the comment. While we did not focus on the more differentiated populations, as we were investigating external stimuli that could modulate Evi1 in the HSPC compartment. Nonetheless, we performed additional experiments to address this question. As outlined in the **Reviewer Fig.3**, we found no significant changes of Evi1^{high} MPP4 (PBS: 10.4±2.4, TPO:7.1±1, p = 0.9), CMP (PBS: 0.6±0.1, TPO:0.5±0.1, p = 0.9), GMP (PBS: 0.03±0.003, TPO:0.5±0.009, p = 0.9) MEP (PBS: 0.09±0.01, TPO:0.1±0.01, p = 0.9) 48h after TPO injection. As no significant differences were found we suggest to simply mention this finding as “data not shown” on **line 134** in the revised MS.

Fig1G does not support the claim that TPO increases Evi1+ HSPCs (no significance and 2 PBS replicates with great variability). A more accurate and robust way to measure this population would be flow cytometry.

While the quantification of the whole-bone imaging did not reach significance, they confirm and nicely illustrate our findings by flow cytometry (**Fig.1C-D**). Due to the significant mouse-to-mouse variability, we imaged additional 2 TPO-treated Evi1-GFP mice which showed similar increase as shown in the revised **Fig.1G**.

Fig10: The decrease in Mpl expression upon TPO injection is an expected result? It seems counterintuitive and some explanation to this would be helpful.

We thank the reviewer for his question. Several older studies have shown that upon binding of TPO the MPL receptor gets internalized and degraded which is part of a classical autoregulatory mechanism (PMID:8555478, PMID:7742532, reviewed in <https://doi.org/10.1056/nejm199809103391107>). We implemented the latter reference into the paper (**line 169 of the revised MS, Ref.29**).

- **Fig2A&B:** a reference to this type of colony classification according to their invasiveness phenotype is missing.

We thank the reviewer for the comment. The colony classification was established in our previous work (<https://doi.org/10.1016/j.ccell.2016.05.011>) based on earlier work by others (<https://doi.org/10.1016/j.stem.2008.11.015>) GMP-derived expand to form mostly compact type I colonies, while LT-HSC and MPP1 can form colonies with a “grape-to scar-like” morphology not resembling any of the classical colony types, therefore referred to as “type IV”. Type IV colonies express lower levels of Fc γ RII/III than cells generally do in type I and type III colonies. Dispersed type IV colonies grown at low density maintained their type IV growth characteristic but also converted to a type I phenotype. In contrast, type I colony-forming cells propagated only as type I colonies. To clarify this point, we show representative images of each colony type in a novel **Supplementary Figure 2A**.

Also, the Evihigh MPP-derived blasts resulting from TPO injection look more aggressive and numerous than the LT-HSC derived. Could it be that panels have been mislabeled?

We thank the reviewer for this interesting observation. However, even though the cytoplots show more cells from the MPP1-derived colonies, the overall number was fairly similar between LT-HSC- and MPP1-derived cells. To not mislead any readers, we replaced the cytoplot picture (representing a different area) of the cells derived from MPP1-colonies (**Fig.2B in the revised MS**).

Fig2C&D are misnamed in the manuscript (line 157).

Due to the addition of a schematic (**Fig.2D**) requested by another reviewer, these figures are now **Fig.2C** and **Fig.2E** and correctly named.

Fig1D is missing the baseline survival of iKMT2A-MLLT3 AML.

Fig.1D does not show any survival. The reviewer most likely talks about Fig.2D. We are not fully sure what the reviewer means with “baseline survival”. Here we compared only mice developing leukemia upon transplant with different cells. As a further comparison we could add survival of DOX-induced iKMT2A-MLLT3 mice (as shown in our previous work (<https://doi.org/10.1016/j.ccell.2016.05.011>)). However, by doing so, we would compare different things. Likewise, if we would add survival of mice transplanted with similar numbers of LT-HSC (also from our previous work) it is not directly comparable as the isolation scheme have significantly changed and improved.

The data **on 2E and 2F** do not show splenomegaly.

We apologize for not being clear. In fact the spleens were significantly enlarged in all groups that are compared (LT-HSCEV^{high}, MPP1Ev^{jhigh}, LT-HSCTPO⁺, MPP1TPO⁺) which is visible by the grey area in **Fig. 2F** (representing the average size of spleens from healthy mice based on <https://phenome.jax.org/>), and also clearly stated in the respective figure legend.

Suppl. Fig2C-D could use larger magnification and marking the leukemic blasts with arrows.

We apologize for not making this clearer. We think that the chosen magnification of the original Suppl. Fig.2C is necessary to show the extent of the leukemic infiltration, however to clarify and to fulfill the reviewer's request, we increased the size of the pictures and added arrows clearly showing the (**revised Suppl. Fig.2D**) organ infiltration by leukemic blasts. In addition, as requested we increased the magnification of the smears in Suppl. Fig.2D would help to see the details better (**revised Suppl. Fig.2E**).

Increase in Gr1*CD11b+ cells is a hallmark of MLL-AF9 AML, what is the relevance of an increase of Fc γ RII/III and why is this important in the characterization of the leukemic phenotype?

Previous work has show that KMT2A-MLLT3 leukemic stem cells express of myeloid lineage-specific antigens that are downstream of the normal progenitor compartment like Fc-gammaRII/III (<https://doi.org/10.1016/j.ccr.2006.08.020>) which was confirmed by our own earlier work (<https://doi.org/10.1016/j.ccell.2016.05.011>).

Data shown in Fig 2H do not appear to be strong enough (25% over half of the recipients, when the control -PBS- shows 1/3). Also, it's not clear what the X axis in the plot is depicting.

We thank the reviewer for his comment. We think that the reviewer was misled by the color of the +TPO (blue) vs. -TPO (red). To clarify this, we switched the colors in revised **Figs. 2I**) This plot shows that mice that received TPO develop more EVI1^{high} AML than those treated with PBS. In addition, we labeled the X axis indicating "symptomatic mice over time". We also tried to statistically evaluate these differences by Wilcoxon test comparing % of Evi1-GFP+ cells between the TPO and PBS group Even though they do not reach significance (p=0.075) there is a clear trend to higher Evi1+ GFP cells in AML derived from TPO-stimulated donors.

Fig2I& J are missing the correlation coefficient (line 172 states a p-value).

We thank the reviewer for his sharp eyes, and apologize for oversee this. We now added the correlation coefficients and statistics to the revised MS (Lines 197-198) and figures (Fig.2J&K)

Fig3 What is the rationale to exclude Flt3+ LSK iKMT2A-MLLT3 cells?

The rationale to exclude Flt3+LSK is based on our observation that the Evi1-expressing cells were in the Flt3- fraction (outlined below). Moreover, Francesco Camargo and colleagues also genetically ablated *Flt3* to enrich for Evi1^{high} LT-HSC by ablating Flt3 (<https://doi.org/10.1038/s41586-020-1971-z>).

This experiment is missing a fundamental t=0 control.

We apologize not to be clear about that. Although it could be interesting to compare the effects of TPO after day-2 and day-5 to a normal control), our goal was to compare effects at day-2 and day-5 with and without TPO exposure. Therefore, we sequenced the samples in batches separated for day-2 and day-5 which resulted in significant batch-effect between samples, rendering a direct comparison of day-2 vs. day-5 or vs. day 0 impossible.

Annotation of the populations' cell identity is missing and ADT data are not shown. What is the rational for performing CITEseq in this experiment?

The rationale for using CITE-seq was to be able to specifically define LT-HSC and MPP1 population in the scRNA experiments. However, contrary to our expectations, it was difficult to distinguish CD34⁺ from CD34⁻ cells (see below) most likely due to limitations of the CD34 antibody (also seen by other groups, pers. communications).

However, to clarify the cells annotations as asked by the reviewer, we replaced the Figs.3C & 3D in the revised manuscript with a t-SNE showing the annotated cells based on transcriptomic signatures found in literature (new **Fig.3B**) and the cell annotations based on the ADT (new **Fig.3C**), and thus, these figures allow to observe the low percentage of LT-HSC cell population such as its t-SNE distribution compared to MPP population.

A) t-SNE dimensionality reduction plot, with color illustrating the cell surface protein enrichment measured for CD34, CD150 and CD48 in CITE-seq.

B) Cell density (y-axis) calculated according to the ADT enrichment measured for CD34, CD150 and CD48 in CITE-seq.

Administration of DEG 5 days following Dox seems to have similar effect as its administration 2 days following Dox. Increase of Cd74 and Socs2 is mentioned, as well as of Pbx3, Prmt6, Fos and Junb, but these data are not shown in the figure.

To clarify, **Figs.3E-F** show changes in gene expression in ST-HSC and MPP-1 after 2 days of DOX: here we discuss Cd74 and Socs2 as well as Cd52 that are all shown in the Vulcan plots. **Fig.3I-J** show changes in gene expression in LT.HSC and MPP2, after 5 days of DOX: here we discuss Pbx3, Prmt6, Fos and Junb are all shown in the -Vulcan plots.

The figure also lacks p-values for the GSEA signatures on **panels H and I**.

We thank for this comment. FDR<=5% was added to **Fig.3G-H** in the revised MS.

Fig4.- It is unclear what panel B represents. What does norm cpm represent and how was it calculated?

To clarify, **Fig.4B** compares different gene expression signatures from iKMT2A-MLLT3 AML from this study (“noTPO_bulk”, “noTPO_Evi1”, “TPO_bulk”, “TPO_Evi1+”) with signatures from our previous work in which we addressed the cellular origin of the disease. Hereby AML originating from LT-HSC that developed the disease early (“LT-early”) or later (“LT-late”) were characterized by expression of *Mecom* and *Erg* mRNA while GMP-derived disease was *Mecom/Erg-low* (<https://doi.org/10.1016/j.ccell.2016.05.011>). We also changed the label into “CPM”, as “count per million” is naturally normalized to the total number of reads.

Erg expression in relation to Evi1 expression is shown but there is no rationale for introducing Erg in this analysis. In addition, there does not seem to be any clear differences among any groups in this panel.

The rationale to introduce Erg is based on our previous finding that Erg and Evi1 expression levels are dissecting AML cells depending on their cellular origin: LT-HSC-derived AML is Evi1^{high}Erg^{high}, while GMP-derived disease is Evi1^{low}Erg^{low} (<https://doi.org/10.1016/j.ccell.2016.05.011>). This observation was confirmed and extended by two more recent studies, indicating that Evi1 directly induces Erg expression (<https://doi.org/10.1182/bloodadvances.2022008018>, <https://doi.org/10.1182/blood.2022016592>). To clarify this for the readers, we added some words in the Text of the revised MS (**Lines 242-244**).

Fig 4G lacks normalized enrichment scores and FDR/ p-values. See also the suppl.Fig3 => same problem

We thank the reviewer for spotting this mistake. Figures have been amended to include the enrichment scores and the p-values (**Fig.4G&H; & Suppl.Fig.4F&G**).

Fig5.- The authors focus on TARGET, the smallest database of the 4 shown in **panel 5C**, to validate the expression data obtained with the mouse model.

Based on the reviewer's comments we apparently were not clear enough that we analyzed *not only* the TARGET database (as illustrated in **Fig.5A-C**) but also the LEUCEGENE, ST.JUDE and BEAT databases (shown in **Suppl. Fig.5A-C**) clearly stated in the MS (**lines 265-267**).

Panel 5D: pearson coefficients show a modest correlation between MECOM and IL12R β 2/INPP4B. The data for the other databases shown in **Suppl. Fig.5**, which are larger in size, show a weaker correlation.

We appreciate the reviewer's comment. To address this point, we performed a more detailed analysis by splitting the patients based on their molecular features (KMT2A-MLLT3, KMT2A-r, MECOM-r or others) and explored TARGET, ST. JUDE, BEAT and LEUCEGENE. We found that the correlations were significant for the extended TARGET and ST-JUDE datasets that mostly comprise pediatric patients. In contrast we observed no correlation between MECOM and INPP4B in patients with KMT2A-MLLT3. This data is now shown in a **revised Fig.5 (D&E**, showing TARGET data) and **Suppl. Fig. 5C** data for the other cohorts. We also comment the findings in the **revised MS (lines 291-295)**.

- **Fig6.-** For functional validation, silencing of MECOM and its targets is performed in the OCI-AML4 cell line because it expresses high levels of MECOM MPL, IL12R β 2 etc. They show that silencing decreases colony forming units and cell growth in vitro. However, these events are happening outside the context of the iKMT2A-MLLT3 fusion and in in vitro and therefore they do not prove relevance of the pathway to the

disease induced by the fission. To answer this question shRNA knockdown of MECOM should be performed in the mouse model that expresses the iKMT2A-MLLT3 fusion. Effects on AML burden, and survival should be assessed in vivo.

We outlined this point in the second paragraph to the reviewer in details.

Supp.Fig.5 shows a western blot that is missing normalization and quantification, as well as controls for the nuclear and cytoplasmic compartments.

As requested by the reviewer we added the quantification of the Western blot bands as well as the image of the Ponceau-stained blot membrane (**Suppl. Fig.6A-C**).

Reviewer #3 - AML, mouse models (Remarks to the Author):

In this manuscript, Chatel-Soulet et al seek to clarify the cell of origin of EVI1 expressing KMT2Ar AML using a novel murine model with both inducible expression of the KMT2A-MLLT3 oncofusion and an EVI1 reporter allele. Using this model system they report that administration of exogenous TPO results in a greater fraction of cycling LT-HSC and MPPs and a greater propensity for malignant transformation of these stem/progenitor cells by KMT2A-MLLT3. Gene expression analysis of the resultant murine AML as well as human AML RNAseq data sets identified differential expression of several stem cell associated genes. Knock down of one identified gene, IL12Rb2 (in addition to knock down of MECOM) led to impaired growth of murine and human EVI1+ KMT2Ar AML cells. Overall, the study is well done. The animal modelling is elegant with nice orthogonal validation of some of their findings using human data and a relevant human AML cell line. However a number of issues must be addressed before the manuscript is suitable for publication.

1. Do the authors think the enhanced transforming capacity of HSCs/MPPs after TPO exposure is specific to KMT2A-rearrangement? It seems there are many human AMLs with high EVI1 expression lacking a KMT2Ar, would they postulate other leukemic drivers may likewise have increased ability to transform activated cells after TPO exposure?

The reviewer addresses an interesting point. MECOM expression has been recognized as an important independent adverse prognostic factor in AML. Notably, elevated MECOM expression was also shown to be a prognostic marker of poor outcome in a subgroup of KMT2A-r AML (Groschel, S., *et al.* J Clin Oncol **31**, 95-103, 2013). Interrogation of 2 datasets containing mostly pediatric patients (TARGET, ST. JUDE) and 2 datasets of mostly adult patients (BEAT, LEUCEGENE) indicates that high levels of MECOM expression is not limited to KMT2A-rearranged AML. Indeed, MECOM^{high}/ERG^{high} (cluster 4, light blue) AML from the TARGET (A) and STJUDE (B) cohorts comprises KMT2A-r as well as patients with NUP98 fusions, mutant NPM1 or other lesions (see the **Reviewer Figure** below).

A**B**
This data again suggests that in absence of any structural 3q26 lesions, high level of *MECOM* expression most likely simply reflects the disease origin in an HSC very high up in the hematopoietic hierarchy rather than being a target of the respective driver lesions. This could mean, as indicated by the reviewer, that our findings may also be valid of EVI1+ AML driven by other lesions. We implemented this point in the revised discussion of the MS (lines 436-438).

2. The mouse experiments and results are poorly described throughout the text making it hard for the reader to follow at times.

Following the reviewer's point, we added two novel schematics explaining the mouse experiments shown in novel **Fig.2D** and **Suppl.Fig.1A** and modified the respective texts (**lines 108-110 and 179-182**).

a. It is not always clear what comparisons are being made or if these are significant or not. For example, they state in **lines 108-110** that TPO led to increased LT-HSC, WBC and trend towards increase BM cellularity, but in suppl fig 1 the WBC and BM cellularity look to be no different than PBS, so are the saying increased compared to the other treatments?

We thank the reviewer for this well-taken point. As the Suppl. Fig.1 shows changes for all treatments, we also added the respective values (%) and p-values also into the text. We show that TPO results in an increase of LT-HSC, while no significant changes in WBC, PLT and BM cellularity were found.

In lines 123-124 it is stated that MPP1 (compared to LT-HSCs) have a 'higher S/G2/M fraction'. While the fraction of MPP1 in S/G2/M is listed, it is not for LT-HSC and based on the figure it also appears these fractions are not significantly different, which is counter to the way it is written in the text.

We apologize for not being clear. We revised this sentence and also provide the respective numbers (**lines 129-134 of the revised MS**). As we aim to focus on Evi1+ cells we moved this figure that shows the global changes into the supplementary data file as a **new Suppl. Fig. 1I**.

The text states in lines 124-127 that the cycling fraction of LT-HSC and MPP1s increase with TPO, but based on **fig 1I** it appears this increase is not significant. Such overstatements must be avoided and if the findings are not significantly different should be stated as such.

Again we apologize not to be precise enough. The striking finding here is that we observed only increased cycling of Evi1-positive (Evi1^{high}- more than Evi1^{low}) LT-HSC,

while Evi1^{neg} cells were more in G₀ (Fig.1I). These very selective changes are not well visible in the global view in the old Fig.1J which we therefore moved to the **Suppl. Fig.1I**.

b. Why in **figure J** and **1K** are the fraction of S/G2/M of both untreated LT-HSC and MPP1s so much higher than that shown in **1I**? Were these done under different conditions?

We apologize for not been clear enough. The old Fig1I showed the proportion (%) of all LT-HSC and MPP1 cells in different phases of the cell cycle. In contrast the old Fig.1J and Fig.1K further dissected these cells into Evi1^{high}, Evi1^{low} and Evi1^{neg} fractions. As our old version appeared not clear enough, we now reordered the figure: we moved the old Fig.1I to the supplementary data set (**new Suppl. Fig.1I**), and show LT-HSC +/- TPO split into Evi-high-low-negative fractions (**revised Fig.1H-I**), followed by the MPP1 in the same format (**revised Fig.1J-K**).

c. In **figure 2I and J**, counter to the text, it appears there is a correlation between EVI1 expression and disease latency in the mice not stimulated with TPO. What statistical test was applied? The number in the formula for the -TPO is odd ($y = -0.7.442 * X_{86.38}$), what is -0.7.442?

Here the statistical test applied was the Pearson correlation test. The reviewer is right, -0.7.442 cannot be a correct: we corrected it to -0.7442 and added p-values and removed the formula (**revised Fig.2J-K**).

d. The timing of doxycycline induction of KMT2A-MLLT3 relative to the injections of TPO, plpC and 5-FU is not clear in the first three result sections. The results section starting at line 146, is particularly difficult discern the experiments that are being performed. A description at the beginning of these sections of the experiments being conducted and an experimental schema (like is shown in Fig 3A) is needed.

We thank the reviewer for the comment. We added additional schematics (**new Fig.2D, & new Suppl. Fig.1A**) for more clarity in the description of our mouse experiments.

3. In **figure 2G**, it is a bit surprising that more of the MPP1 mice with or without TPO are EVI1+ compared to the LT-HSC even with TPO. Can the authors propose some reason for this finding?

We thank the reviewer for the comment. The susceptibility of more committed cells such as GMP for malignant transformation by retrovirally expressed KMT2A-MLLT3 has been functionally linked to active cell cycle progression (Chen, X *et al.*, Nat Commun 10, 5767, 2019). LT-HSC has been described as a compartment composed of mostly quiescent cells (which we confirm by flow cytometry; **Fig.1I**), while MPP1 have a higher proportion of already cycling cells (**Fig.1K**). Thus, we believe that the fusion is able to initiate the transformation more easily in the MPP1, “freezing” the status of the cell of origin which express Evi1.

4. Why was the differential gene expression analysis of patient data done comparing EVI1^{high} to EVI1 intermediate whereas the mouse data was EVI1+ vs EVI1^{neg}? Do the identified gene expression differences replicate if in human data EVI1+ is compared to EVI1^{neg}?

Thank you for this comment. In order to address this point, we also investigated the differential gene expression between the MECOM^{high} and MECOM^{low} patients from all AML cohorts (TARGET, ST.JUDE, LEUCEGENE, BEAT) (see below). Hereby we observed that comparison between MECOM^{high} vs MECOM^{intermediate} resulted in less DEGs than comparison of MECOM^{high} vs. MECOM^{low/neg} (with the exception of BEAT). Importantly, *MECOM* and *IL12Rb2* were found in all comparisons, while *INPP4* was not significant in the ST.JUDE and TARGET datasets.

Minor issues:

There are a few typos:

a) In line 310-311 it says “While induction of of *INPP4B*-targeting shRNAs resulted in cell death (data not shown), it decreased growth and colony formation in MOLM13 cells.” Presumably it resulted in cell death in OCI AML4 cells? Please this sentence.

We thank the reviewer for the correction. It is indeed OCI-AML4 that were very sensitive upon shRNA-mediated knockdown of *INPP4B*. This has been corrected in the revised MS (line 346-347).

b) In line 314 it says ‘...and expressed higher expression levels...’. Remove either expressed or expression from this sentence.

Thank you for spotting this. We corrected is in the revised MS (line 3)

c) In line 349 it states ‘..activation of STAT5 activation..’ remove one activation from this sentence.

This has been corrected in the revised MS (line 394)

d) Please define all abbreviations at first use including KME in line 116 and MC in line 150.

Thank you for the reminder. This has been corrected in the revised MS (lines 84 &176)

There are also issues with the figures that should be addressed:

a) Figure 1 B – in the text (line 113) it indicates that MPP2 and MPP3 are shown in Figure 1B, but Figure 1B shows MPP1. Please fix the text or the figure.

Thank you for the correction. MPP2 and MPP3 are indeed shown in the supplementary figures (**Suppl.Fig.1E&F**). The mistake has been corrected in the revised MS (**lines 117 & 121-122**)

b) Some of the text is far too small to read. In figure 1 C and D the EVI1 high/low/neg are almost indiscernible. Likewise for EVI1+ and EVI1- in figure 4A, B and E. Same for the labels for the bar graph color schema in Suppl Fig 2H-J and all the font for figure Suppl Fig 4E and F.

We appreciate the input of the reviewer on this point. We increased the font size for the following figures: **Fig.1.C-D & O, Fig.2.A-C & E-H, Fig.4.A-D, Suppl Fig.1.G-H & L-N, Suppl Fig.2.I-J, Suppl Fig.4.A-B and Suppl Fig.5.E-F** in order to render them more legible.

c) There is no label for the color scheme in the bar plots for Suppl Fig 1F and G.

We added the legend in the revised supplementary figure (**Suppl.Fig.1G&H**)

d) It appears in the text in lines 246 and 252, figures 5A and 5B, respectively are erroneously called out as 6A and 6B.

Correction to 5A and 5B has been done in the revised MS (**lines 274 & 280**)

Reply to Reviewer 2

1. We appreciate the reviewers point regarding the GESA dataset performed with a custom TPO-signaling expression signature deduces from reference no. 44 not reaching significance ($p=0.098$). Further interrogation of the MSig-DB revealed a BioCarta TPO-pathway expression signature that resulted in a more significant GSEA comparison ($p=0.009$). We therefore now show this GESA as new Fig.4H. However, although not reaching statistical significance, the comparison with the custom signature is based on a much higher number of genes ($n=96$) than the BioCarta-derived signature ($n=41$), we therefore think that the first comparison is most likely as biologically relevant and therefore we also show it as Suppl. Fig. 4H.

2. As suggested by the reviewer, we performed the trajectory analysis using the scanpy (v1.11.3) python (v3.12.11) package, accessed through the reticulate (v1.41.0.1) interface in R (v4.4.1) of the scRNA seq datasets shown in Fig.3 of our MS. The dataset was first split by time-point (day 2 and day 5) and only cells from the myeloid clusters were retained. These were further subdivided into four experimental conditions: PBS with KMT2A-MLLT3 (+DOX), PBS without KMT2A-MLLT3 (-DOX), TPO with KMT2A-MLLT3 (+DOX), and TPO without KMT2A-MLLT3 (-DOX).

The neighborhood graph was computed using scanpy.pp.neighbors, based on the batch-corrected PCA space ($nc=30$) with 5 nearest neighbors. PAGA analysis (scanpy.tl.paga) was then performed using the clustering defined in the original analysis. Diffusion maps (scanpy.tl.diffmap) and pseudotime inference (scanpy.tl.dpt) were calculated using the LT-HSC cluster as the root.

The PAGA connectivity graph was converted into an undirected, weighted graph using the igraph package (v2.1.4). For ease of comparison with previous visualizations, nodes were manually positioned to match the t-SNE layout by placing each one at the centroid of its corresponding cluster. Edges with weights below 0.2 were excluded from the final plots to retain only meaningful connections.

We summarized the findings in a Figure for the Reviewer. In panel A, we compare trajectories derived from cells collected after 2 and 5 days on DOX (expressing the KMT2A-MLLT3 fusion), without (left) and with (right) TPO — reflecting the comparisons shown in Fig. 3D–G (day 2) and H–K (day 5). Panel B shows the same comparison for cells analyzed after 2 and 5 days off DOX (not expressing the fusion), again without (left) and with (right) TPO. Across all comparisons, the overall structure of the inferred trajectories was largely preserved, with only minor differences in the connectivity between clusters. These subtle variations are consistent with the low number of differentially expressed genes observed at these time points.

Figure for Reviewer-2: